# ANGEL2 phosphatase activity is required for non-canonical mitochondrial RNA processing

Paula Clemente [1] ✉, Javier Calvo-Garrido[1], Sarah F. Pearce[1,9], Florian A. Schober [1,10], Megumi Shigematsu[2], Stefan J. Siira [3,4], Isabelle Laine [1], Henrik Spåhr[1], Christian Steinmetzger[1], Katja Petzold [1], Yohei Kirino [2], Rolf Wibom[1,5], Oliver Rackham[3,4,6,7], Aleksandra Filipovska [3,4,6], Joanna Rorbach [1,8], Christoph Freyer [1,5,11] ✉ & Anna Wredenberg [1,5,11] ✉

Canonical RNA processing in mammalian mitochondria is defined by tRNAs acting as recognition sites for nucleases to release flanking transcripts. The relevant factors, their structures, and mechanism are well described, but not all mitochondrial transcripts are punctuated by tRNAs, and their mode of processing has remained unsolved. Using *Drosophila* and mouse models, we demonstrate that non-canonical processing results in the formation of 3′ phosphates, and that phosphatase activity by the carbon catabolite repressor 4 domain-containing family member ANGEL2 is required for their hydrolysis. Furthermore, our data suggest that members of the FAST kinase domain-containing protein family are responsible for these 3′ phosphates. Our results therefore propose a mechanism for non-canonical RNA processing in metazoan mitochondria, by identifying the role of ANGEL2.

Transcription of the mammalian mitochondrial genome (mtDNA) is initiated from two designated promoter regions, which generate long, primary polycistronic transcripts that cover almost the entire length of the mtDNA molecule[1]. The mitochondrial tRNA punctuation model, proposed 40 years ago, defines that tRNAs intersperse the mitochondrial genome and act as excision sites to release the individual transcripts prior to maturation[2]. Since then, the responsible ribonucleases RNase P and ELAC2 and their mechanistic and structural basis have been described[3–9]. Mammalian mtDNA encodes for 2 rRNA, 22 tRNAs and 11 mRNAs, and this model explains processing of most

transcripts in metazoan mitochondria[7,9–13]. However, in mammals at least two gene junctions do not follow this canonical pattern, and the mechanism of their processing has remained mainly unknown. Several members of the FAST kinase domain containing protein family have been associated with cleavage of such junctions[14–17]. Specifically, FASTKD4 and FASTKD5 have been implicated in the maturation of transcripts lacking flanking tRNAs, but the mechanism is unclear[18].

After excision, the individual transcripts are further modified by polyadenylation of most mRNAs by the mitochondrial polynucleotide adenylyltransferase MTPAP[19–22], and methylation and pseudouridylation

[1]Department of Medical Biochemistry and Biophysics, Karolinska Institutet, 171 65 Stockholm, Sweden. [2]Computational Medicine Center, Sidney Kimmel Medical College, Thomas Jefferson University, Philadelphia, PA, USA. [3]Harry Perkins Institute of Medical Research, QEII Medical Centre and University of Western Australia, Nedlands, WA 6009, Australia. [4]ARC Centre of Excellence in Synthetic Biology, QEII Medical Centre and University of Western Australia, Nedlands, WA 6009, Australia. [5]Centre for Inherited Metabolic Diseases, Karolinska University Hospital, 171 76 Stockholm, Sweden. [6]Curtin Medical School and Curtin Health Innovation Research Institute, Curtin University, Bentley, WA 6102, Australia. [7]Telethon Kids Institute, Northern Entrance, Perth Children's Hospital, 15 Hospital Avenue, Nedlands, WA, Australia. [8]Max Planck Institute Biology of Ageing—Karolinska Institutet Laboratory, Karolinska Institutet, 171 65 Stockholm, Sweden. [9]Present address: Simons Initiative for the Developing Brain, University of Edinburgh, Edinburgh, UK. [10]Present address: Max Planck Institute of Biochemistry, 82152 Planegg/Martinsried, Germany. [11]These authors contributed equally: Christoph Freyer, Anna Wredenberg. ✉e-mail: paula.clemente@ki.se; christoph.freyer@ki.se; anna.wredenberg@ki.se

of mitochondrial tRNAs and rRNAs[23–35]. While the mechanisms of transcription, processing, and translation are increasingly understood[36,37], the role of polyadenylation inside mitochondria is less clear, and has been proposed to act as both degradation and stabilisation signal[20]. Several factors have been shown to directly affect MTPAP activity and polyadenylation. For instance, the polynucleotide phosphorylase, PNPase, and the ATP-dependent RNA helicase SUV3 have been shown to inversely regulate poly(A) tail length via MTPAP[38–44]. Furthermore, the leucine rich pentatricopeptide repeat containing protein, LRPPRC, stabilises mitochondrial mRNAs and stimulates MTPAP activity[41,45,46].

While several mitochondrial transcripts require polyadenylation to complete their translational stop codons, *mt:Nd6* is the only mRNA not polyadenylated in humans or mouse for yet unknown reasons[20]. The role of the poly(A) tail in translation is not clear, but we previously demonstrated that transcripts lacking a poly(A) signal can still be translated but lose their 3′ integrity[21]. Previous studies revealed that polyadenylation occurs prior to full partitioning of the primary transcripts[10,47], necessitating a deadenylase to remove such tails from immature tRNAs prior to maturation, and although a PNPase/SUV3 complex has been shown to degrade mRNAs[38–42], their role in deadenylation is not clear. In mitochondria only the carbon catabolite repression 4 (Ccr4) family member, 2′,5′-phosphodiesterase, PDE12, has been proposed to have deadenylation function[48–50], capable of removing spurious poly(A) additions on selected mitochondrial tRNAs[50]. But whether additional deadenylases function within mitochondria is unknown.

Here, we identify that ANGEL2 and its *Drosophila* homologue, *Dm*Angel, are mitochondrial proteins that are essential during non-canonical processing. Using *Drosophila* and mouse knock out models, we demonstrate that upon loss of *Dm*ANGEL or ANGEL2, transcripts that undergo non-canonical processing, accumulate 3′ phosphates, preventing their polyadenylation, causing a respiratory chain deficiency. Through immunoprecipitation experiments, we show that *Dm*Angel interacts with a member of the FASTK protein family, and its deletion prevents cleavage during non-canonical processing. Our results describe the mechanism of non-canonical processing and identify the formation of terminal phosphates as an important intermediary step.

## Results

### CCR4-family Angel proteins share evolutionary history with the deadenylase PDE12 and are localised to mitochondria

We set out to identify additional factors involved in the deadenylation of mitochondrial transcripts and thus performed a biological pathway analysis based on evolutionary deduction[51], using both MTPAP and PDE12 as bait. While MTPAP returned no candidates, PDE12 was co-evolutionarily absent with other Ccr4 family members in some amoeba and protozoa (Supplementary Fig. 1a). All three factors carry an endonuclease/exonuclease/phosphatase domain (PFAM: PF03372), thought to be involved in the removal of the polyadenylation signal of mRNAs due to its strong conservation among known deadenylases[52]. CCR4C, also known as Nocturnin (NOCT) affects cytosolic poly(A) tail dynamics during the circadian rhythm in mice[53] and recently has been linked to inter-conversion of the dinucleotides NADP⁺/NADPH and NAD⁺/NADH[54]. The paralogs CCR4E and CCR4D, also known as ANGEL1 and 2, resulted from a gene duplication in the vertebrate lineage with only a single ANGEL homologue present in *Drosophila* (CG12273)[55,56]. ANGEL1 has previously been described to localise to the endoplasmic reticulum and/or Golgi apparatus and interact with the translation initiation factor 4E, eIF4E[57], while ANGEL2 has recently been shown to hydrolyse terminal 2′,3′-cyclic phosphates (2′,3′cP) on RNA molecules in vitro[58].

In silico prediction models suggested a mitochondrial localisation for both human ANGELs (*Hs*ANGEL1 and *Hs*ANGEL2), as well as for *Drosophila* ANGEL (*Dm*ANGEL) (Supplementary Fig. 1b). We confirmed

this by expressing C-terminally tagged GFP-fusion proteins in human fibroblasts (Fig. 1a), and subcellular fractionation experiments (Fig. 1b and Supplementary Fig. 1c). We next performed proteinase K, swelling, and sodium carbonate treatments to determine both sub-mitochondrial localisation and membrane association of *Hs*ANGEL1, *Hs*ANGEL2, and *Dm*ANGEL (Supplementary Fig. 1d, e). Together, our results are consistent with an outer mitochondrial membrane localisation of ANGEL1, while ANGEL2 and *Dm*ANGEL localise to the mitochondrial matrix (Fig. 1c), suggesting a functional homology between ANGEL2 and *Dm*ANGEL.

### Deletion of DmAngel in flies and ANGEL2 in mice causes respiratory chain deficiency

To understand their physiological roles in the mitochondrial matrix, we studied CRISPR/Cas9 knockout models in both fly (*DmAngel^KO*) and mouse (*MmAngel2^KO*) (Supplementary Fig. 2a–d). Loss of *Dm*ANGEL resulted in a lethal phenotype during the pupal stage, with no homozygous KO flies hatching (Supplementary Fig. 2e). Correct targeting was confirmed by expressing a *DmAngel* transgene (*DmAngel^Rescue*) with or without a FLAG-tag in the *DmAngel^KO* background, rescuing the lethal phenotype (Supplementary Fig. 2f–h). Furthermore, expression of a transgene encoding a catalytically dead mutant *Dm*ANGEL protein (*DmAngel^E121A*)[48] failed to rescue the *DmAngel^KO* phenotype, validating that *Dm*ANGEL activity is necessary for normal fly development (Supplementary Fig. 2h). Deletion of *Angel2* in mice (*MmAngel2^KO*) resulted in viable offspring at Mendelian proportions, without any obvious phenotype until 15 months of age, when we stopped observing the animals.

We next measured mitochondrial respiratory chain enzyme activities, and *DmAngel^KO* larvae presented with a mitochondrial dysfunction in complexes I, IV and V (Fig. 1d and Supplementary Fig. 2i), while an isolated complex I deficiency was observed in heart mitochondria from 16-week-old *MmAngel2^KO* mice (Fig. 1e and Supplementary Fig. 2j). Untargeted proteomic analysis in *DmAngel^KO* larvae (Fig. 1f and Supplementary Data File 1) and *MmAngel2^KO* hearts (Fig. 1g and Supplementary Fig. 2k and Supplementary Data File 1) confirmed that the mitochondrial dysfunction most likely is caused by reduced OXPHOS subunit levels. Together, these results suggest that *Dm*ANGEL and ANGEL2 have a role in mitochondrial gene expression.

### Hydrolysis of 3′ terminal phosphates by DmANGEL and ANGEL2 is required during non-canonical processing

We measured mRNA steady-state levels in both models, revealing that *mt:Atp8/6* and *mt:Nd5* in mouse and *mt:nd5*, *mt:atp8/6-mt:co3* and *mt:nd6-mt:cytb* in fly (Fig. 2a–g and Supplementary Fig. 3a–c) were significantly upregulated, while mt:tRNA levels were mostly unchanged in both model systems (Supplementary Fig. 3d–i). An exception are *mt:tRNA^Cys* and *mt:tRNA^Tyr* in *Drosophila*, which were both significantly reduced. Notably, while the gene content of the *Drosophila* and mammalian mtDNA is identical, the gene order differs (Fig. 2d, h) and the affected messenger transcripts all require non-canonical processing (i.e. not separated by tRNAs) in *MmAngel2^KO* or *DmAngel^KO* samples, with the exception of *mt:nd5* in fly (Fig. 2e–g). Furthermore, Northern blot analysis revealed that the *mt:Atp8/6* transcript migrated slightly faster in *MmAngel2^KO* samples, in comparison to controls (Fig. 2a). We therefore measured mitochondrial poly(A) tail length (MPAT) (see materials and methods). Most transcripts showed a normal polyadenylation pattern in hearts of 16-week-old *MmAngel2^KO* mice and *DmAngel^KO* larvae (Fig. 3a, b and Supplementary Fig. 4a, b). In contrast, *mt:Nd5* and *mt:Atp8/6* in mouse, and *mt:nd6* and *mt:atp8/6* transcripts in fly presented with a reduced polyadenylation pattern in *MmAngel2^KO* and *DmAngel^KO* samples, respectively (Fig. 3a, b).

We argued that since the 3′ ends of these transcripts undergo non-canonical processing, *Dm*ANGEL and mammalian ANGEL2 could partake in their polyadenylation. Polyadenylation requires a 3′ hydroxyl to

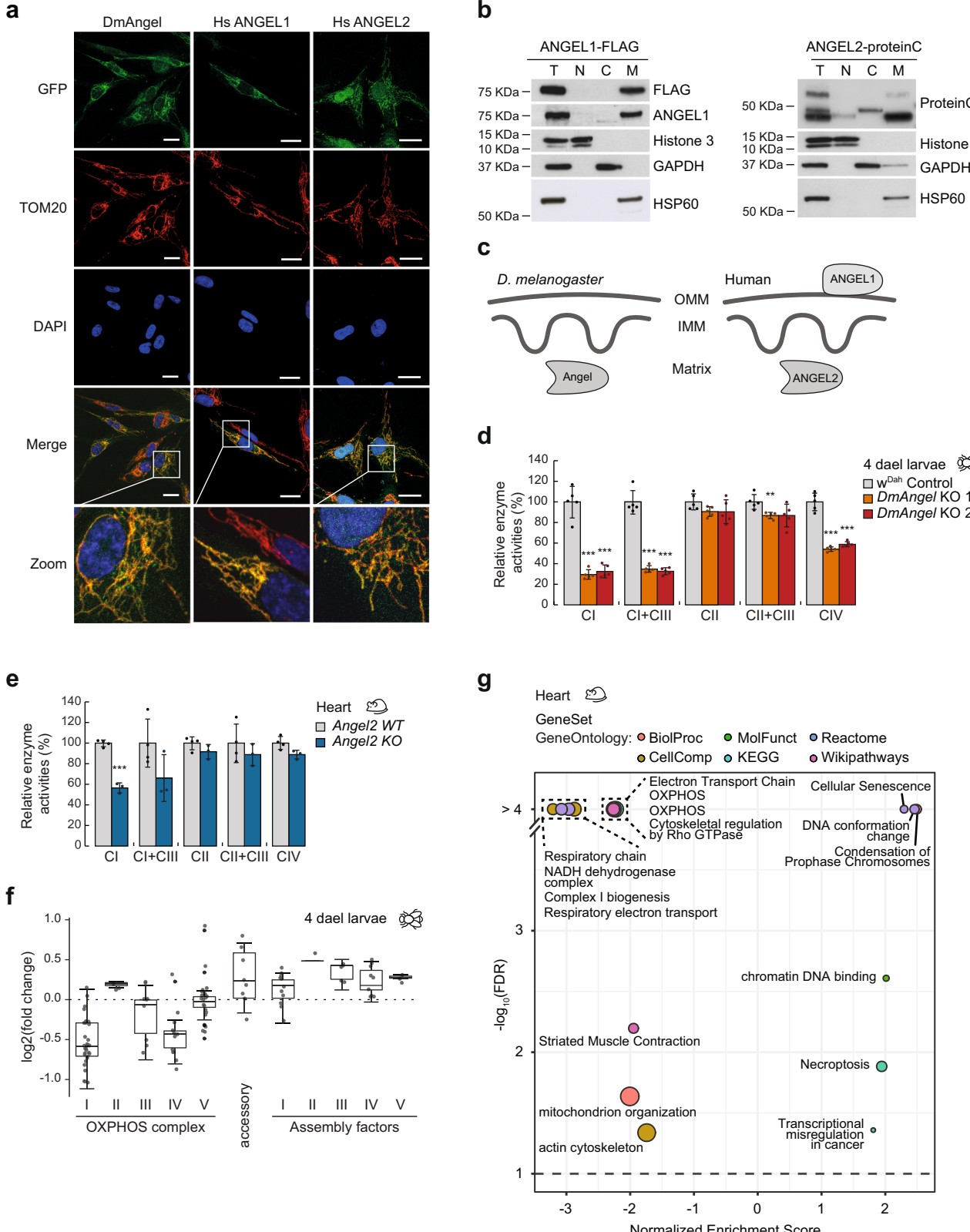

ligate incoming adenosine residues to the terminal residue, and nucleases usually hydrolyse phosphodiester bonds to retain a free 3′ OH. However, for some ribonucleases the scissile bond is downstream of the phosphorus atom, resulting in a 3′ phosphate end and a free 5′ hydroxyl upon cleavage. Additionally, free 2′ phosphates or 2′,3′-cPs have also been reported[59]. The MPAT assay relies on ligating primers to the 3′ end of the transcripts prior to amplification and separation on

high-resolution polyacrylamide gels and therefore relies on the presence of 3′ hydroxyls. To test whether cleavage during non-canonical processing retains a terminal phosphate, we pre-treated RNA samples with either calf-intestinal phosphatase (CIP), which hydrolyses phosphomonoester bonds but cannot hydrolyse 2′,3′-cP, or polynucleotide kinase (PNK), which can hydrolyse all three terminal phosphates (Supplementary Fig. 4c). Both treatments revealed large amounts of

**Fig. 1 | DmANGEL and MmANGEL2 are mitochondrial proteins required for normal respiratory chain function. a** Confocal imaging of human skin fibroblasts expressing GFP-tagged *Dm*ANGEL, *Hs*ANGEL1 or *Hs*ANGEL2 constructs. (Scale bar = 20 mm; Zoom = 4.5X). Representative images are shown of three independent experiments with 20 cells analysed per experiment. **b** Western blot analysis of sub-cellular fractionations from human skin fibroblasts expressing ANGEL1-FLAG or ANGEL2-proteinC fusion proteins expressed in human skin fibroblasts, decorated with antibodies against nuclear (Histone 3), cytosolic (GAPDH) or mitochondrial (HSP60) fractions. Representative experiment of two independent experiments, with two technical replicas. **c** Diagram depicting the sub-mitochondrial localisations of *Dm*ANGEL, ANGEL1 and ANGEL2. Mitochondria are illustrated as outer mitochondrial membrane (OMM), inner mitochondrial membrane (IMM) and matrix. Isolated respiratory chain enzyme activities for NADH:ubiquinone oxidoreductase (CI), NADH:cytochrome *c* oxidoreductase (CI + III), succinate:ubiquinone

oxidoreductase (CII), succinate: cytochrome *c* oxidoreductase (CII + III), and cytochrome *c* oxidase (CIV), in isolated mitochondria from (**d**) 4-days-after-egg-laying (dael) *Dm*ANGEL KO (red; *DmAngel KO1,2*) and control (light grey; w$^{Dah}$) larvae, and (**e**) 16-week-old mouse hearts (blue; *Angel2 KO*) and controls (light grey; *Angel2 WT*). (Data in **d**, **e** are represented as mean ± SD; \*$p < 0.05$, \*\*$p < 0.01$, \*\*\*$p < 0.001$ with two-tailed Student's *t* test; Dm $n = 5$, Mm wt $n = 4$, Mm KO $n = 3$ biologically independent samples. **f** Proteomic levels of individual subunits of the five OXPHOS complexes, accessory, and OXPHOS assembly factors in 4-dael *DmAngel$^{KO}$* larvae. Data are presented as 25 to 75 percentile box with median, and whiskers represent ±1.5× inter-quartile range ($n = 4$ biologically independent samples). No statistical test was performed. **g** Gene set enrichment analysis of proteomic data in 16-week-old mouse hearts, relative to controls. *p* values are given as false discovery rates after multiple testing adjustment ($n = 3$ biologically independent samples).

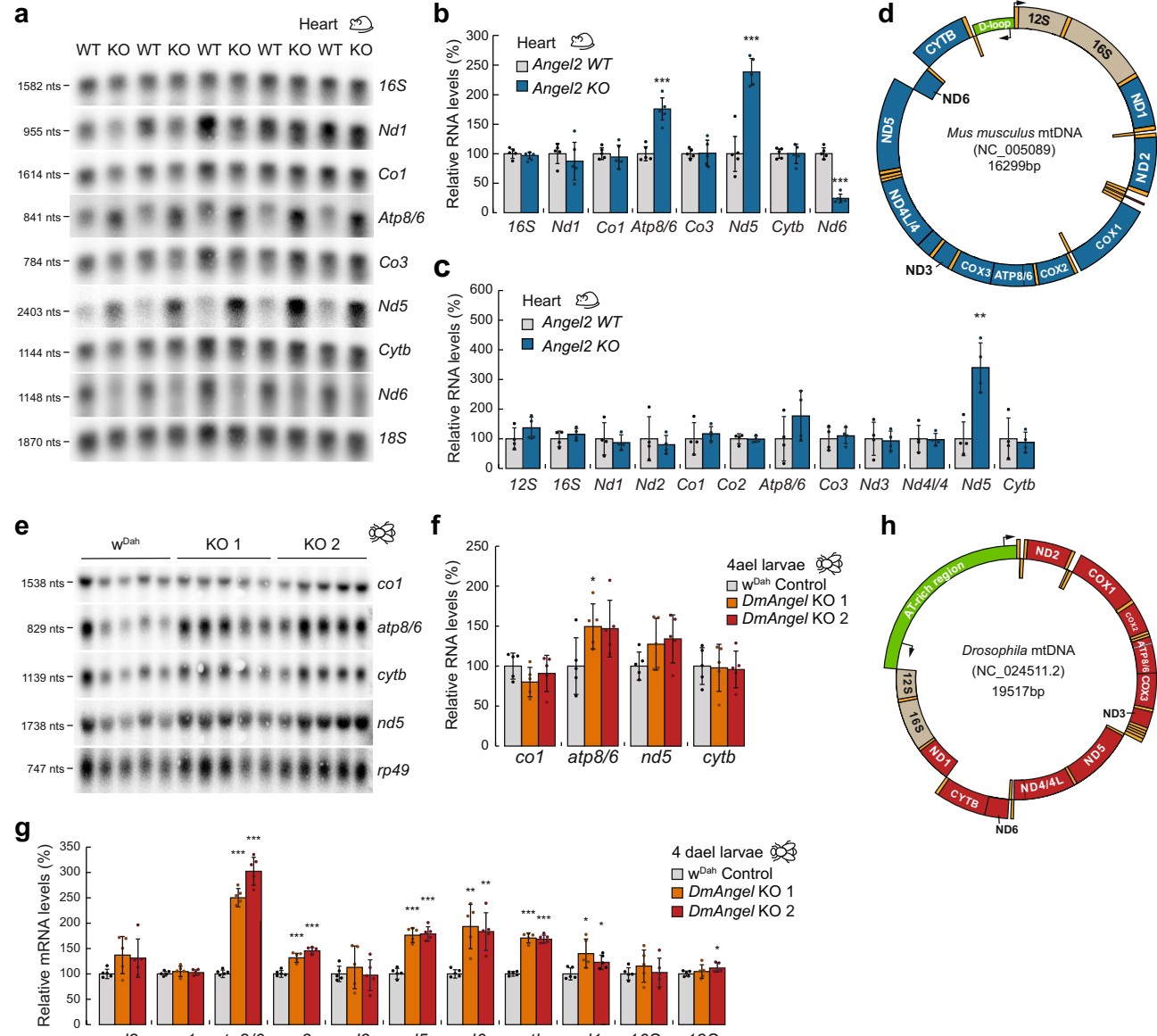

**Fig. 2 | Deletion of DmAngel or MmANGEL2 affects mitochondrial gene expression.** Relative mitochondrial transcript steady-state levels in 16-week-old mouse hearts (blue; *Angel2 KO*) and controls (light grey; *Angel2 WT*), as determined by Northern blot analysis ($n = 5$ biologically independent samples) (**a**, **b**) or qRT-PCR ($n = 4$ biologically independent samples with 3 technical replicas) (**c**). 18S rRNA was used as loading control. **d** Illustration of mouse mtDNA (NC_005089). Black arrows depict transcription initiation sites and tRNAs are indicated in orange.

Relative mitochondrial transcript steady-state levels in 4-dael *Dm*ANGEL KO (red; *DmAngel KO1,2*) and control (light grey; w$^{Dah}$) larvae, as determined by Northern blot analysis (**e**, **f**) or qRT-PCR (**g**). RP49 was used as loading control ($n = 5$ biologically independent samples). **h** Illustration of *Dm* mtDNA (NC_001709). Black arrows depict transcription initiation sites and tRNAs are indicated in orange. (All data are represented as mean ± SD; \*$p < 0.05$, \*\*$p < 0.01$, \*\*\*$p < 0.001$ with two-tailed Student's *t* test).

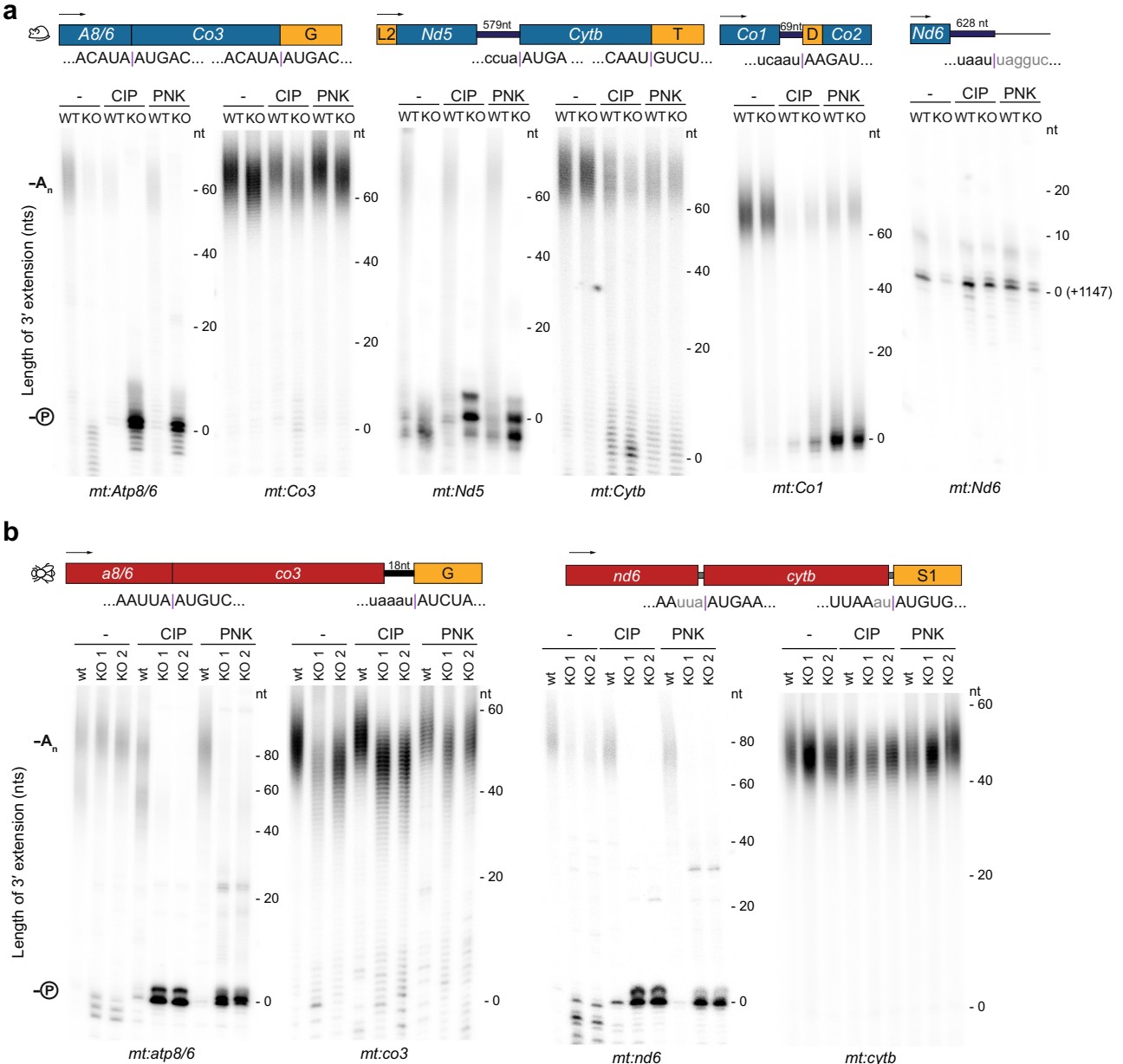

**Fig. 3 | Non-canonical processing of mitochondrial gene junctions results in the formation of 3′ phosphates that are hydrolysed by DmANGEL or MmANGEL2.** **a** Mitochondrial polyadenylation tail (MPAT) length assay in *MmAngel2^KO* mouse heart samples, performed with or without CIP or PNK pre-treatment, as indicated. **b** Mitochondrial polyadenylation tail (MPAT) length assay in *DmAngel^KO* fly samples, performed with or without CIP or PNK pre-treatment, as indicated. CIP calf intestinal phosphatase, hydrolyses phosphomonoester bonds. PNK T4 polynucleotide kinase, hydrolyses phosphomonoester bonds or 2′,3′ cyclic phosphodiester from RNA ends. Gene junctions and their sequences are indicated. Non-coding sequences are lower case. Sequences not annotated are shown in lower case grey. tRNAs are shown in orange with their single letter code. Poly(A) tails (-A$_n$) and 3′ phosphates (-℗) are shown. A representative experiment is shown of at least three independent experiments performed with biologically independent samples.

non-polyadenylated 3′ termini on *mt:Nd5* and *mt:Atp8/6* in *MmAngel2^KO* mice, and *mt:nd6* and *mt:atp8/6* in *DmAngel^KO* larvae, due to the presence of a terminal phosphate group (Fig. 3a, b, -℗). In contrast, their downstream transcripts *mt:Cytb* and *mt:Co3*, showed normal polyadenylation with no effect of CIP or PNK treatment. The same observations were made in liver, skeletal muscle, and kidney samples of *MmAngel2^KO* mice, suggesting that ANGEL2 activity is ubiquitous (Supplementary Fig. 5a–c).

Ligation of an RNA linker using RtcB ligase, which requires free 3′ phosphates or 2′,3′-cP, followed by MPAT, further confirmed the presence of 3′ phosphates on *mt:Nd5* and *mt:Atp6* in mouse and *mt:nd6* and *mt:atp8/6* in fly (Fig. 4a, b). Our results therefore suggest that ANGEL2 and *Dm*ANGEL hydrolyse 3′ phosphates formed during non-

canonical processing in vivo. To confirm phosphatase activity and to demonstrate that *Dm*ANGEL and *Mm*ANGEL are indeed orthologs, we incubated recombinant *Dm*ANGEL and an N-terminally truncated *Mm*ANGEL2 with oligonucleotides containing 2′-, 3′-, or 2′,3′-cPs. In agreement with observations made with human ANGEL2[58], both ANGEL homologues efficiently hydrolysed 2′- or 2′,3′-cPs, but not 3′ phosphates in vitro (Fig. 4c). This discrepancy with our in vivo data suggests DmANGEL/ANGEL2 activity differs in vivo, or that the physiological conditions in the absence of the ANGEL homologues favour 3′ phosphate formation after cleavage.

CIP or PNK treatment also revealed a prominent pool of non-polyadenylated transcripts for mouse *mt:Co1* transcripts independent of *MmANGEL2* (Fig. 3a). Murine *mt:Co1* contains a short 3′ UTR

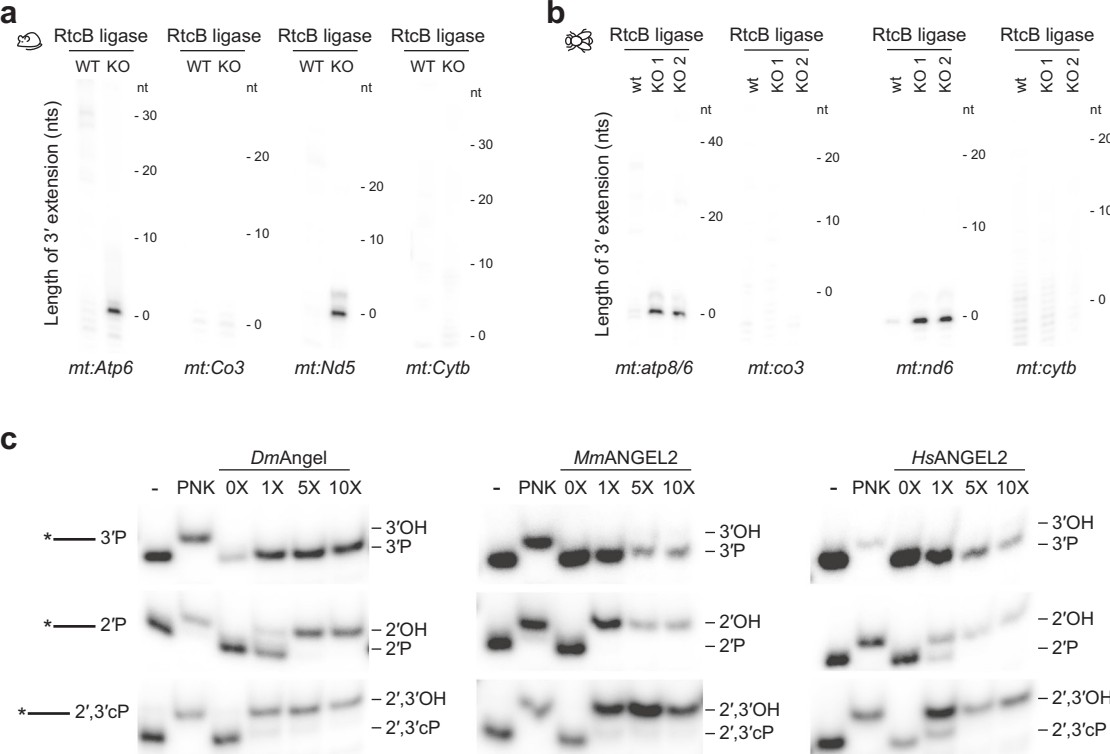

**Fig. 4 | ANGEL2 and DmANGEL are terminal phosphatases in vivo and in vitro.** MPAT length assay using RtcB ligase in **a** *MmAngel2*[KO] mouse heart or **b** *DmAngel*[KO] fly samples. **c** In vitro phosphatase activity of recombinant *Drosophila* (*Dm*Angel), mouse (*Mm*ANGEL), or human (*Hs*ANGEL) ANGEL or ANGEL2 protein, in the presence of RNA oligos (20 nts) containing 3'-, 2'-, or 2',3'-cPs as indicated. PNK treatment was used as positive control. Representative experiments are shown of at least two independent experiments performed with biologically independent samples.

corresponding to anti-tRNA[Ser1] and has previously been considered to undergo non-canonical processing at its 5' end[14,15]. Additionally, a non-polyadenylated pool of *mt:Co1* transcripts has previously been observed[60]. We tested whether the other member of the CCR4 family, NOCT, was involved in de-phosphorylation of *mt:Co1*. NOCT has recently been suggested to have a mitochondrial-localised isoform[54], but we observed no difference in polyadenylation or 3' phosphate status in samples from NOCT KO mice[61] (Supplementary Fig. 6a). Finally, mammalian *mt:Nd6* is the only mitochondrial mRNA not polyadenylated and has been reported to be notoriously difficult to clone[20]. However, our MPAT analysis does not support that a free 3' phosphate prevents *mt:Nd6* polyadenylation (Fig. 3a), and maps its 3' end to position +1147, with a weaker signal at +1148. In contrast, we did observe a weak oligoadenylation signal, which we confirmed by sequencing (Supplementary Fig. 6b), suggesting that some *mt:Nd6* transcripts indeed have short poly(A) tails.

### Loss of Angel affects mitochondrial translation and the translatome composition

The absence of polyadenylation results in an *mt:Atp6* transcript lacking a complete translational stop codon[20]. We also observed a mild effect on de novo translation in heart and liver mitochondria from 16-week-old *MmAngel*[KO] mice with ND5 slightly upregulated, probably due to the increased *mt:Nd5* transcript levels (Supplementary Fig. 7). In contrast, *DmAngel*[KO] larvae had a more aberrant translation pattern (Fig. 5a), consistent with the different severity observed in mouse and fly. Sucrose density fractionations of mitochondrial lysates revealed increased mitochondrial monosome formation in the absence of *Dm*ANGEL (Fig. 5b and Supplementary Fig. 8a), suggesting either increased translation or stalling. Mitoribosome profiling[62] revealed increased association of *mt:cytb* mRNA with the translating mitoribosome, and reduced association with *mt:co1*, *mt:co3* and *mt:nd3* mRNAs

(Fig. 5c and Supplementary Fig. 8b, c). This suggests that loss of *DmAngel* alters the translation efficiency of specific mRNAs. Additionally, we found a significant reduction of *mt:tRNA*[Met] associated with the translating mitoribosome in *DmAngel*[KO] larvae, consistent with stalling, rather than increased translation initiation (Fig. 5d). Coverage analysis of the mitoribosomal footprints along *mt:atp6* mRNA revealed increased reads corresponding to the 3' end of the mRNA and lack of polyadenylation required to complete the STOP codon in *DmAngel*[KO] larvae, implying stalling of the ribosome (Supplementary Fig. 8c, d). In contrast, the 3' end of *mt:nd6* was not over-represented in knockout samples, consistent with the presence of a complete STOP codon, and normal release from the mitoribosome.

### DmAngel is part of the processing machinery of non-canonical transcripts

Finally, we performed immunoprecipitation (IP) experiments in combination with stable-isotope labelling of amino acids in fruit flies (SILAF)[63] to identify factors interacting with *Dm*ANGEL. With mass-spectrometry-based proteomics we identified, among others, enrichment for CG13850, a mitochondrial protein of unknown function (Fig. 6a and Supplementary Fig. 9a and Supplementary Data File 1). We confirmed this interaction in a reciprocal IP, using flies expressing a CG13850-FLAG tag fusion protein (Fig. 6b and Supplementary Data File 1). A homology search identified CG13850 as a homologue of the mammalian FAS-activated serine/threonine kinase domain-containing protein family (FASTK). Unlike humans, who encode 6 different FASTK proteins (FASTK, and FASTKD1 to 5)[18], *Drosophila* contain only two homologues, and CG13850 most closely aligns with human FASTKD4 (Fig. 6c), which is supported by its interaction with ANGEL2 in a recent mitochondrial proximity study[64]. Silencing of *CG13850* in flies specifically affected the processing of the non-canonical junctions at *mt:atp8/6*–*mt:co3* and *mt:nd6*–*mt:cytb* (Fig. 6d), strongly supporting the notion

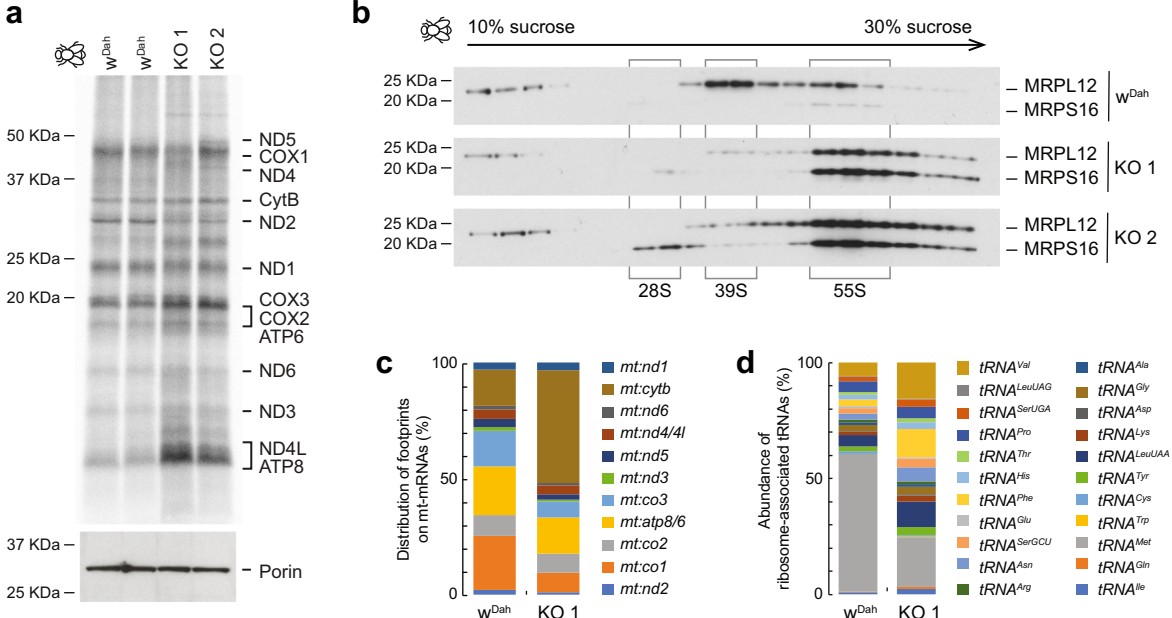

**Fig. 5 | Incomplete non-canonical processing due to loss of DmANGEL results in aberrant mitochondrial translation. a** De novo translation in isolated mitochondria from 4-dael control (w^Dah) and *DmAngel*^KO (KO 1,2) larvae. Mitochondrial proteins are indicated. Porin was used as loading control. A representative experiment is shown with *n* = 2 biologically independent samples, of two independent experiments performed. **b** Western blots of ribosome gradient fractions from 4-dael control and *DmAngel*^KO larvae, decorated with antibodies against the small (MRPS16) and large (MRPL12) mitochondrial ribosome subunits. The small (28S), large (39S), and monosome (55S) fractions are indicated. A representative experiment is shown of four independent experiments performed with biologically independent samples. **c, d** Abundance of mitoribosome footprints in mitochondrial transcripts from ribosome profiling experiments in control and *DmAngel*^KO as indicated, normalised to total read count and expressed as proportion of total footprints (*n* = 2 biologically independent samples).

that CG13850, and its mammalian homologues FASTKD4 and 5, are responsible for processing of non-canonical junctions. Additionally, knockdown of *CG13580* resulted in reduced mRNA steady-state levels (Supplementary Fig. 9b, c), aberrant mitochondrial translation (Supplementary Fig. 9d), and compromised OXPHOS activity (Fig. 6e), strongly supporting that CG13850 is the *Drosophila* ortholog of members of the FASTK family.

## Discussion

The mitochondrial tRNA punctuation model was proposed 40 years ago and defines the canonical mode of RNA processing, where tRNAs intersperse the mitochondrial genome and act as excision sites to release the individual transcripts prior to maturation[2]. In contrast, non-canonical processing describes the release of transcripts not flanked by tRNAs, but its mechanism is largely unknown. Here we demonstrate that cleavage of such junctions results in unconventional terminal phosphates, requiring phosphatase activity by ANGEL2 in mammals and *Dm*ANGEL in *Drosophila melanogaster* prior to polyadenylation. Using fly and mouse KO models, as well as recombinant *Dm*ANGEL and ANGEL2, we establish that *Dm*ANGEL is the *Drosophila* ortholog of ANGEL2. Interaction studies with *Dm*ANGEL further suggest that non-canonical processing requires members of the FASTK family, which most likely, are responsible for cleavage.

ANGEL2 has previously been suggested to localise to the nucleus, where it forms a deadenylase complex together with hCaf1z (also known as TOE1) in the nuclear Cajal bodies[65] or interact with p21 to control the cell cycle[66]. However, these studies used N-terminally tagged fusion constructs, which interfere with the mitochondrial localisation signal, directing the fusion protein to the wrong cellular compartment. In agreement, neither deadenylation activity nor TOE1 interaction could be confirmed in later studies[58,67,68]. Likewise, we failed to observe any link to the cellular UPR in the proteome of mice deficient of ANGEL2 (Fig. 1g and Supplementary Fig. 2k) as previously proposed[58].

In contrast, we here demonstrate that ANGEL2 and *Dm*ANGEL localise to the mitochondrial matrix, which is in agreement with previous large-scale subcellular proteomic studies[64,69,70]. Furthermore, we establish that unlike NOC, ANGEL2 and *Dm*ANGEL hydrolyse terminal phosphates formed during non-canonical processing. While our in vivo results suggest that non-canonical processing results in 3′ terminal phosphates, experiments using recombinant proteins confirm previous observations that ANGEL2 and *Dm*ANGEL preferentially hydrolyse 2′- or 2′,3′-cPs in vitro[58]. Whether cleavage in vivo results in 2′,3′-cPs, which are successively converted to 3′-P in the absence of *Dm*ANGEL/ANGEL2, or whether the in vivo activity for *Dm*ANGEL/ANGEL2 differs, will require further investigation.

Loss of *Dm*ANGEL is larvae lethal, while *MmAngel2*^KO mice are viable, which might be explained by the reduced gene content of the *Drosophila* genome. While *Drosophila* have only two FASTK homologues, mammals encode 6 different FASTK family members, previously associated with various roles in the post-transcriptional regulation of mitochondrial gene expression[15–18]. All FASTK family members contain a C terminal RAP (RNA binding domain abundant in Apicomplexans) domain that has been suggested to fold into a PD-(D/E)-XK nuclease superfamily fold[16,18], and several studies have linked FASTKD4 and FASTKD5 to non-canonical processing[14–17]. Our results suggest that during non-canonical processing FAST kinases cleave the phosphodiester bond downstream of the bridging phosphorus atom, resulting in free 3′ phosphates and 5′ hydroxyls. This mechanism also fully explains the previous observation that the majority of *mt:Co3* transcripts, which is downstream of *mt:Atp8/6*, contain a 5′ hydroxyl group instead of a monophosphate[71]. In the absence of ANGEL2, the 3′ phosphate groups are retained and transcripts resulting from non-canonical processing cannot proceed to polyadenylation, resulting, in the case of ATP6, in an incomplete STOP codon. It is thus feasible to suggest that *Dm*ANGEL and its interactor CG13850 are required for additional functions in the fly, resulting in the lethal phenotype. Interestingly, loss of *Dm*ANGEL also affected *mt:tRNA*^Cys and *mt:tRNA*^Tyr

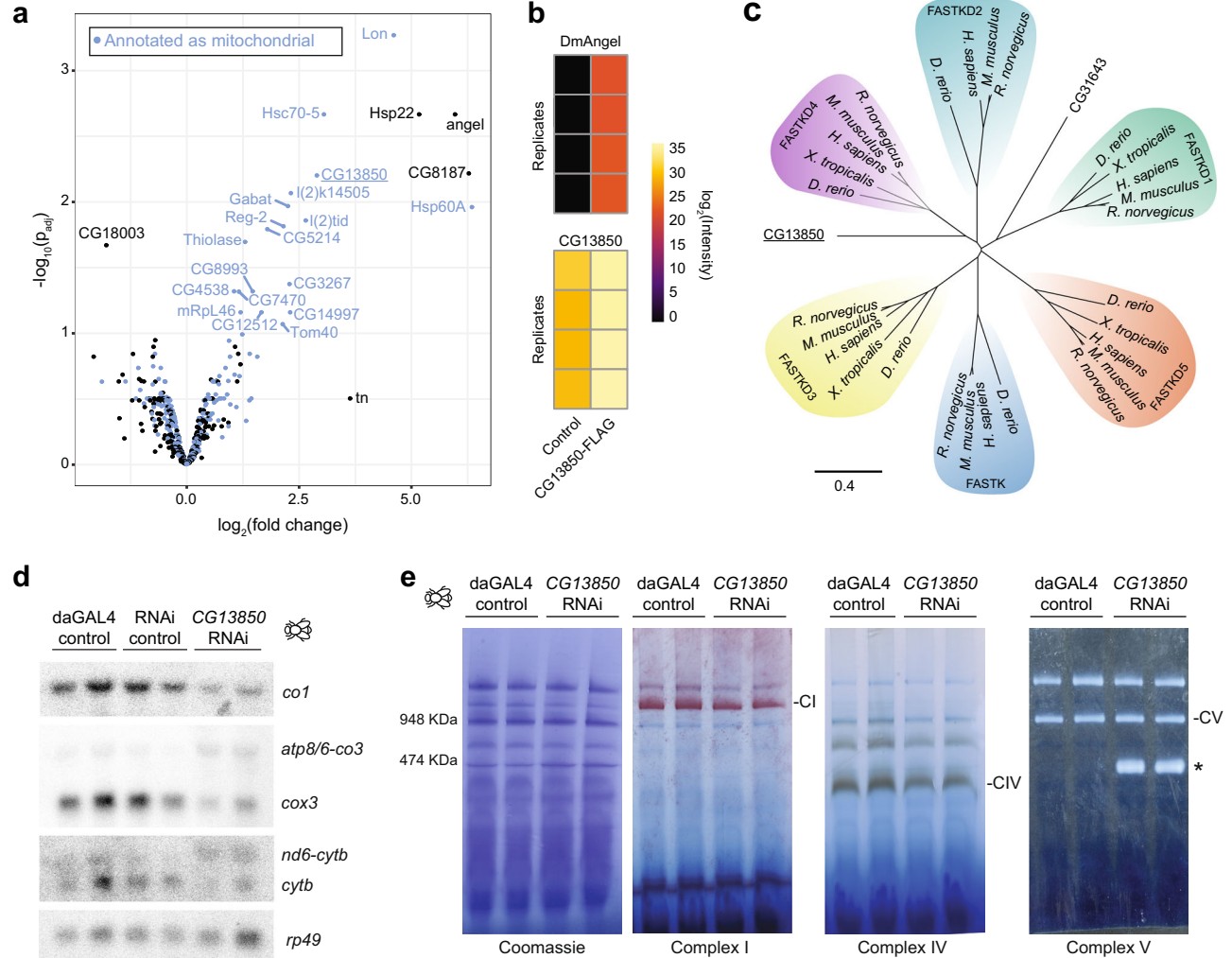

**Fig. 6 | Members of the FASTK family are required for non-canonical processing of mitochondrial transcripts. a** Volcano plot of proteomics data from immunoprecipitation (IP) experiments, using 4 dael larvae from *Dm*ANGEL-FLAG versus *Dm*ANGEL overexpressing larvae (*n* = 4). Proteins in the category "mitochondria" are highlighted in blue. Limma moderated *t*-test, *p* values are given as false discovery rate after adjustment for multiple testing (*n* = 4 biologically independent samples). **b** Heatmap of the enrichment of *Dm*ANGEL and CG13850 in CG13850-FLAG IP experiments (*n* = 4 biologically independent samples). **c** Neighbour-joining protein MUSCLE alignment of 30 FASTK family members (see Supplementary Table 1) from five species (as indicated) shown in an unrouted tree layout. CG13850

and CG31643 are the two *Drosophila* FASTK family orthologs. **d** Northern blot analysis of selected mitochondrial transcripts in 4-dael control (daGAL4 and RNAi) larvae and larvae with silenced *CG13850* (*CG13850* RNAi). RP49 was used as loading control. Representative experiment is shown with *n* = 2 biologically independent samples, of two independent experiments performed. **e** BN-PAGE and in-gel activity staining of NADH dehydrogenase (Complex I), cytochrome *c* oxidase (Complex IV), and ATP synthase (Complex V) with a Coomassie stain as a loading control. Dissociation of the ATPase F$_1$ subunit is indicated by *. Representative experiment is shown with *n* = 2 biologically independent samples.

steady-state levels, which we previously observed in other fly models affecting mitochondrial gene expression[21,41], and which might additionally affect mitochondrial translation. However, the impact on translation, especially in mice, is mild, suggesting that even in the absence of a STOP codon and poly(A) tail, sufficient translation can still occur. This agrees with our previous observation that in the absence of mitochondrial polyadenylation, translation is retained, but mRNA integrity is compromised[21].

The role of polyadenylation of mitochondrial transcripts is not fully understood, although oligoadenylation, i.e. the addition of only a few adenine residues, is required to complete several translational stop codons[20]. Interestingly, we observed substantial differences in poly(A) tail length of several transcripts in kidney and liver, versus heart and muscle mitochondria. The reason for this difference between tissues is not clear to us but might expose additional functions of the poly(A) tail in mitochondrial gene expression.

Together, our results are in full agreement with the notion that ANGEL2 is responsible for the dephosphorylation of the 3′ ends of mitochondrial transcripts and defines a fundamental step of non-canonical processing in mitochondria (Fig. 7).

## Methods

### Generation of a DmAngel knock-out line

The generation of the *DmAngel* knock-out lines (*DmAngel*[KO#1] and *DmAngel*[KO#2]) was performed as previously described[72]. The sequence of the *DmAngel* gene, obtained from sequencing genomic DNA from the fly line nos-Cas9 (Bloomington Stock #54591), was entered into CRISPR Optimal Target Finder (http://targetfinder.flycrispr.neuro.brown.edu) and the target sites with the lowest off-target score were selected (GCAGATTTCTTTCCTCCACTCGG and GGTTGTCTCATATAACATCCTGG). The gRNAs were cloned into BbsI digested pCFD4 (Addgene plasmid #49411) by Gibson assembly, as detailed in[72].

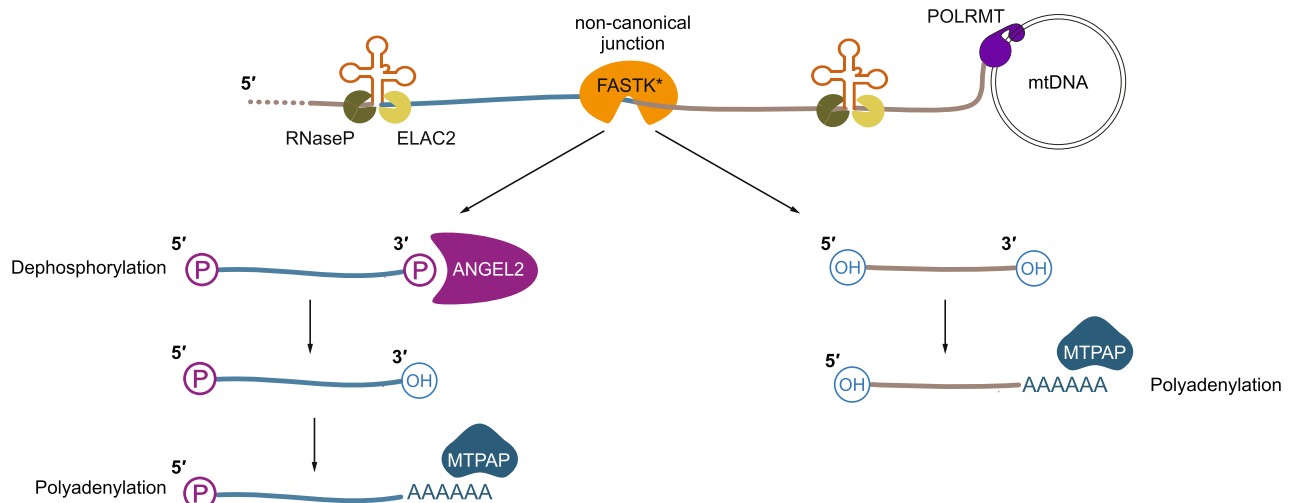

**Fig. 7 | ANGEL2 phosphatase activity is required for the maturation of mRNAs that undergo non-canonical processing.** Model of the non-canonical processing of mitochondrial RNA. The mitochondrial RNA polymerase, POLRMT, transcribes long, polycistronic transcripts, which are cleaved by RNase P and ELAC2 at the tRNA gene junctions. Non-canonical junctions are recognised by members of the FASTK family (FASTK*), resulting in the formation of 3′ phosphates. Phosphatase activity by ANGEL2 hydrolyses the 3′ ends of *mt:Nd5* and *mt:Atp8/6*, allowing further their polyadenylation by the mitochondrial poly(A) polymerase, MTPAP.

Primers used to generate the constructs are detailed in Supplementary Data File 2. The pCFD4 plasmid containing the gRNAs was used to generate a gRNA expressing fly line, using BestGene Inc (California, USA).

Males carrying the gRNAs were crossed to virgin nos-Cas9 females for germline-specific editing. Individual offspring flies were crossed to CyO/Gla, Bc balancer flies and the progeny was individually screened for deletions in the *DmAngel* gene as followed. The wings of candidate flies were clipped and incubated at 37 °C for 45 min in freshly prepared adult fly homogenisation buffer (10 mM Tris-HCl, pH 8.2, 25 mM NaCl, 1 mM EDTA, 0,2 μg/μl proteinase K) and the homogenate was subsequently used for PCR screening and sequencing[73]. Successful *DmAngel*[KO] lines were backcrossed for 6 generations to a w;; background to remove possible off-target effects.

### Generation of genomic DmAngel^rescue, DmAngel^rescue-FLAG, and DmAngel^E121A flies

The genomic *DmAngel* locus (6.5Kb from BAC clone CH322-6L23, BACPAC Resource Centre, California, USA) was cloned by ET recombination into the pBluescript II SK + vector (Stratagene), modified for FLAG tag addition or E121A mutation, and further subcloned into pattB plasmid. Transgenic lines were generated via P-element-mediated germ line transformation (BestGene Inc).

### Generation of DmAngel, DmAngel^FLAG, CG13850, and CG13850^FLAG overexpression fly lines for immunoprecipitation

*DmAngel* and *CG13850* cDNAs were PCR amplified with or without an in frame STOP codon and flanking attB sites for cloning into pDONR201 (Thermo Fisher Scientific), before subcloning into pTWF (*Drosophila* Genomics Resource Center), and finally into pUASTattB with or without a FLAG tag at their 3′ ends. The overexpressing lines were generated via P-element-mediated germ line transformation, by injecting pUASTattB-*DmAngel*, pUASTattB-*DmAngel*-FLAG, pUASTattB-*CG13850*, or pUASTattB-*CG13850*-FLAG plasmids into *Drosophila* embryos using BestGene Inc. Primers used to generate the constructs are detailed in Supplementary Data File 2.

### Drosophila stocks and maintenance

For in vivo knockdown of *CG13850* a w;UAS-*CG13850*RNAi; line (#15233) was obtained from Vienna *Drosophila* Resource Center (VDRC). Ubiquitous down-regulation was achieved by crossing the UAS-RNAi lines to the driver *daughterless*GAL4 (w;;daGAL4). Experimental samples are labelled as follows throughout the text: daGAL4 Control (w;;daGAL4/+, daGAL4 line crossed to w;;), RNAi Control (w;UAS-RNAi/+;, RNAi line crossed to w;;), RNAi (w;UAS-RNAi/+;daGAL4/+, RNAi line crossed to daGAL4).

All fly stocks were backcrossed for at least 6 generations into the Wolbachia-free white Dahomey background (w;;). All fly lines were maintained at 25 °C and 60% humidity on a 12 h:12 h light:dark cycle on a standard yeast-sugar-agar medium.

### ANGEL2 knock out mice and mouse husbandry

*Angel2* KO mice were obtained from Nanjing Biomedical Research Institute, Nanjing University, China (Strain number: T001889; B6/N-Angel2^emICd32/Nju) and were backcrossed to the C57BL/6N strain (Charles River Laboratories). Mice were housed in standard individually ventilated cages with a 12-h:12-h light:dark cycle under controlled environmental conditions. Mice were fed ad libitum with standard CRM mouse food (Special Diet Services). All animal procedures were approved by the Stockholm's Animal Experimentation Ethics Board (Ethical permit number 14460/18) of the Swedish Board of Agriculture and were performed in accordance with national and European law. Genotyping primers of *Angel2*[KO] mice are detailed in Supplementary Data File 2.

### Cell culture, nucleofection and transduction

Immortalised human skin fibroblasts (HSFs) were cultured in high-glucose DMEM (Thermo Fisher Scientific) supplemented with 10% foetal bovine serum (Thermo Fisher Scientific) at 37 °C in a 5% $CO_2$ atmosphere. Experiments were performed with cells cultured to 80–90% confluency. Medium was changed regularly, and cells were passaged using TrypLE Express (Thermo Fisher Scientific).

To analyse subcellular localisation, the cDNA for human *ANGEL1*, *ANGEL2* and *DmAngel* were cloned into pDONR201 (Thermo Fisher Scientific) and further subcloned into Gateway pcDNA-DEST47 (Thermo Fisher Scientific), that carries an in-frame GFP gene. Nucleofection of HSFs was performed using a 4D-Nucleofector System (Lonza) and buffer P2, following manufacturer's instructions. 48 h after the nucleofection, fibroblasts were processed for experiments.

Stable expression of human *ANGEL1*, *ANGEL1*-FLAG, *ANGEL2* and *ANGEL2-protein C* in immortalised HSFs was achieved by cloning into

pBABE-Puro-gateway. For retroviral particle generation, Phoenix-AMPHO cells were transfected with the pBABE constructs using Lipofectamine 3000 (Thermo Fisher Scientific) following manufacturer's instructions. Medium was harvested and filtered 48 h after transfection. HSFs were transduced in a 50–50 mixture of fresh and retroviral positive medium with 4 µg/ml polybrene (Merck). Puromycin selection was initiated 48 h after transduction. Primers used to generate the constructs are detailed in Supplementary Data File 2.

## Immunostaining and confocal microscopy
Immortalised HSFs were cultured on 1.5 mm cover slips and fixed for 20 min at room temperature using 3% paraformaldehyde. Permeabilisation and blocking was performed using a combination of 0.1% saponin (Merck), 1% BSA (Merck) in PBS for 1 h. Primary antibody anti-TOM20 (42406, Cell Signalling) was diluted 1:200 in PBS with 0.1% saponine and 1% BSA and incubated overnight. After washing, cells were incubated for 1 h with secondary antibody Alexa Fluor 568 (Goat anti-Mouse IgG (H + L) antibody Alexa Fluor 568, A-11031, Thermo Fisher, or Goat anti-Rabbit IgG (H + L) antibody Alexa Fluor 568, A-11036, Thermo Fisher) at a 1:1000 dilution. Cells were mounted with ProLong Gold Antifade Mountant and 0.01 µg/ml DAPI (Merck). Images were acquired using a Nikon Ti-E inverted point scanning confocal microscope A1R Si.

## Isolation of mitochondria from cell lines, mouse tissues and Drosophila larvae
Immortalised HSFs were collected by scrapping from 8–12 p150 plates and washed with PBS. The pellet was resuspended in mitochondrial isolation buffer (220 mM mannitol, 70 mM sucrose, 1 mM EDTA, 20 mM HEPES pH 7.6) supplemented with 2 mg/ml of BSA and incubated on ice for 15 min. Cells were homogenised by hand with 20 strokes and centrifuged at $800 \times g$ for 5 min at 4 °C. The supernatant was subsequently centrifuged at $10,000 \times g$ for 10 min at 4 °C to obtain the crude mitochondrial fraction.

Mitochondria were isolated from mouse tissues using differential centrifugation as previously described[74]. Briefly, fresh tissue was cut, washed with PBS, and homogenised in a potter homogeniser. Heart tissue was homogenised in 225 mM sucrose, 1 mM EGTA, 20 mM Tris-HCl pH 7.2 and 200 µl of 2.5% trypsin. Liver was homogenised in 310 mM sucrose, 10 mM Tris-HCl, and 1 mM EDTA. The homogenates were centrifuged at $1000 \times g$ for 10 min at 4 °C to pellet nuclei and cell debris and the supernatant was subsequently spun at $10,000 \times g$ for 15 min at 4 °C to obtain the crude mitochondrial fraction.

Isolation of mitochondria from larvae was performed as previously described[45]. Third-instar larvae were washed and homogenised in 250 mM sucrose, 2 mM EGTA and 5 mM Tris pH 7.4 with 1% BSA, nuclei and cell debris were pelleted by centrifugation at $1000 \times g$ for 5 min at 4 °C. The supernatant was centrifuged at $7000 \times g$ for 10 min at 4 °C to obtain the crude mitochondrial fraction.

## Proteinase K protection assays
In total, 250 µg of Drosophila mitochondria were resuspended in 2 mM Tris-HCl pH 7.4, 250 mM sucrose, 2 mM EGTA. Swelling was performed by resuspension of 250 µg of mitochondria in 2 mM Tris-HCl pH 7.4, 2 mM EGTA and pipetting up and down. To disrupt mitochondrial membranes, mitochondria were resuspended in 2 mM Tris-HCl pH 7.4, 250 mM sucrose, 2 mM EGTA and 0.3% Triton X-100. Digestion of accessible proteins was achieved by addition of 31.5 µg/ml or 63 µg/ml proteinase K and incubation for 20 min on ice. Digestion was stopped by addition of a protease inhibitor cocktail. Proteins were pelleted with sodium deoxycholate and TCA and loaded onto 4–12% SDS-PAGE gel for western blot detection of mitochondrial proteins.

In total, 125 µg of HSF mitochondria were resuspended in 20 mM HEPES pH 7.6, 220 mM mannitol, 70 mM sucrose, 1 mM EDTA. Swelling of the mitochondrial membranes was achieved by resuspending

mitochondria in 20 mM HEPES pH 7.6, 1 mM EDTA and pipetting up and down 10 times. To disrupt mitochondrial membranes, mitochondria were resuspended in 20 mM HEPES pH 7.6, 220 mM mannitol, 70 mM sucrose, 1 mM EDTA and 0.3% Triton X-100. Digestion, precipitation, and detection of proteins was performed as described above.

## Alkaline carbonate extraction of mitochondrial membrane proteins
In total, 125 µg of mitochondria were pelleted by centrifugation and resuspended in 0.1 M $Na_2CO_3$ of the corresponding pH, pipetted up and down and incubated on ice for 30 min. Samples were centrifuged in a TLA55 rotor, at $45,000 \times g$ for 30 min at 4 °C, to separate membrane associated proteins in the pellet and soluble proteins in the supernatant. Proteins in each fraction were precipitated by sodium deoxycholate and TCA and loaded onto 4–12% SDS-PAGE gel for western blot detection of mitochondrial proteins.

## Western blotting
Mitochondria were resuspended in NuPAGE LDS sample buffer, loaded on NuPAGE gels (Thermo Fisher Scientific) and transferred to PVDF membranes (Millipore). Primary antibodies used in this study were anti-DDDDK (ab1257, Abcam, 1:2000), ANGEL1 (sab1303061, Merck, 1:1000), Histone 3 (H0164, Merck, 1:2000), HSP60 (AB1-SPA-807-E, Enzo Life Sciences, 1:1000), Protein C (A00637, GenScript, 1:1000), TOM20 (13929S, Cell Signalling, 1:1000), AIF (ab16501, Chemicon, 1:1000), MRPS16 (HPA054538, Merck, 1:1000), MRPL12 (HPA022853, Merck, 1:1000), TIM22 (14927-1-AP, Protein Tech, 1:1000), Porin (ab128568, Abcam, 1:1000), GAPDH (ab8245, Abcam, 1:1000). Membranes were washed and incubated for 1 h at RT with the appropriate secondary antibody at a 1:4000 dilution. Secondary antibodies used in this study were horseradish peroxidase-conjugated anti-rabbit IgG (NA9340-1ML, Cytiva), anti-mouse IgG (NA9310-1ML, Cytiva) or mouse anti-goat IgG (#sc-2354, Santa Cruz Biotechnology). Western blots were visualised using Clarity Western ECL solution (Bio-Rad), following manufacturer's instructions.

## Blue native polyacrylamide gel electrophoresis (BN-PAGE) and biochemical evaluation of respiratory chain function
BN-PAGE and in-gel staining for complex I, IV and V activities was performed as previously described[21]. Mitochondria isolated from larvae or mouse tissues were resuspended in 250 mM sucrose, 15 mM $K_2HPO_4$, 2 mM MgAc$_2$, 0.5 mM EDTA and 0.5 g/l BSA, pH 7.2, and biochemical activities of the respiratory chain complexes were determined as previously described[75].

## RNA isolation, reverse transcription, and quantitative PCR
RNA was isolated using TRIzol (Thermo Fisher Scientific) and quantified with a Qubit fluorometer (Thermo Fisher Scientific) unless otherwise stated. Reverse transcription for qRT-PCR analysis was performed using High-Capacity cDNA Reverse Transcription Kit (Thermo Fisher Scientific). qRT-PCR was performed on a QuantStudio 6 (Thermo Fisher Scientific) with Taqman probes (Thermo Fisher Scientific) or Platinum SYBR Green qPCR supermix-UDG (Thermo Fisher Scientific) and gene-specific primers. Primers and Taqman probes used for qRT-PCR are detailed in Supplementary Data File 2.

## Northern blot analysis of mitochondrial RNAs
Steady-state levels of mitochondrial transcripts were determined by Northern blot analysis as previously described[21]. RNA samples were separated in 1% MOPS-formaldehyde agarose gels and tRNAs were separated by neutral 10% PAGE gels. After separation, samples were transferred to Hybond-N + membranes (GE Healthcare) and hybridised with randomly [$^{32}$P]-labelled dsDNA probes, [$^{32}$P]-labelled strand-specific RNA probes or with [$^{32}$P]-end labelled oligonucleotide probes.

Membranes were exposed to a PhosphorImager screen, and the signal was quantified using a Typhoon FLA7000 system and ImageQuant TL 8.1 software (GE Healthcare). Primers used to generate dsDNA and oligonucleotide probes have been previously described[45,76].

## Mitochondrial poly(A) tail-length assay (MPAT)

MPAT was performed as previously described[77]. In brief, 0.2–2 μg of total mouse or *Drosophila* RNA was ligated to a linker oligonucleotide at the 3′ end using T4 RNA ligase 1 (New England Biolabs) or RtcB ligase (New England Biolabs) and reverse transcribed using M-MLV reverse transcriptase (Thermo Fisher Scientific) and an oligonucleotide complementary to the linker sequence. Poly(A) tails were PCR amplified using a gene-specific forward primer and the anti-linker oligo, followed by precipitation to remove oligonucleotides and free nucleotides. Labelling was achieved with 5 cycles of a nested PCR on the initial PCR product, using internal end-labelled gene specific oligonucleotides and the anti-linker oligo. PCR products were separated on 6 or 8% high resolution polyacrylamide gels and exposed to PhosphorImager screens. Quick CIP (New England Biolabs) and T4 PNK (New England Biolabs) treatment was performed according to manufacturer's recommendations, followed by purification of the treated RNA using acid phenol and precipitation with sodium acetate before the ligation to the linker oligonucleotide. Oligonucleotide linker and primers used for the MPAT assay are detailed in Supplementary Data File 2.

## Cloning and purification of DmANGEL, HsANGEL2 and MmANGEL2

A codon-optimised (Genscript) DNA construct corresponding to *Dm*ANGEL (amino acids 26–354) was cloned into a Pet24b vector and *Hs*ANGEL2 (amino acids 165–544) or *Mm*ANGEL2 (amino acids 164–544) was cloned into a Fh8-Pet24d vector. *Hs*ANGEL2 and *Mm*ANGEL2 were expressed in Arctic express cells (Agilent) at 16 °C for 48 h in Magic Media (Thermo Fisher Scientific). After lysis, the proteins were purified over a His-Select Ni2+ resin (Sigma-Aldrich) and dialysed against H-0.5 (25 mM Tris pH 7.4, 0.5 mM EDTA, 10% glycerol, 3 mM β-mercaptoethanol, 500 mM NaCl) after the addition of TEV protease at a 1:50 protease:protein ratio. The dialysed proteins were loaded on a second Ni2+ resin for TEV protease removal and the flow through was collected. Further purification was conducted over a HiLoad 16/60 Superdex 200 pg gel filtration column (GE Healthcare) in buffer H-0.5 lacking glycerol with the addition of 2 mM dithiothreitol instead of β-mercaptoethanol. *Dm*ANGEL was expressed and purified analogous to *Hs*ANGEL2 except that no His tag removal was performed.

## RNA synthesis

*Hs*Nd5, *Mm*Nd5 and *Dmnd6* oligos with 2′-terminal phosphate groups were prepared by RNA solid-phase synthesis with standard phosphoramidite methods on an H-8 DNA/RNA/LNA synthesiser (K&A Laborgeräte).

5′-*O*-DMT/2′-*O*-TBDMS-protected 3′-β-cyanoethyl phosphoramidites of $N^6$-benzoyladenosine, $N^4$-acetylcytidine, $N^2$-acetylguanosine and uridine were employed as 70 mM solutions in anhydrous MeCN. The 3′-terminal nucleotide was incorporated using a 5′-*O*-DMT/3′-*O*-TBDMS-protected 2′-β-cyanoethyl phosphoramidite of either $N^4$-benzoylcytidine (*Hs*ND5) or $N^6$-benzoyladenosine (*Mm*Nd5, *Dmnd6*) dissolved in anhydrous MeCN at the same concentration. Installation of 2′-terminal phosphate groups was achieved by using phosphate CPG support (1 μmol). Coupling times were 6 min for standard and 9 min for isomeric phosphoramidites, respectively, with 250 mM ETT in MeCN as the activator.

Cleavage from the solid support and removal of the base-labile protecting was achieved by treatment with 25% aq. NH₃/EtOH 3:1 (1.2 ml) at 55 °C for 4 h.

After lyophilisation, the residue was dissolved on anhydrous DMSO (100 μl). 2′-*O*-silyl deprotection was performed by treatment

with Et₃N·3HF (125 μl) at 65 °C for 2.5 h, followed by addition of aq. NaOAc (25 μl, 3 M, pH 6.5). The crude RNA was desalted by EtOH precipitation and purified by preparative denaturing PAGE (20%). The purity of the isolated RNA was confirmed by analytical PAGE (20%) with SYBR Gold staining.

Preparative denaturing polyacrylamide gels (0.7 × 200 × 200 mm) were cast from 20% acrylamide/bis-acrylamide 19:1 and 8 M urea. Gels were run for 2.5 h at 35 W constant power in 1x TBE. The RNA was visualised by UV shadowing, extracted from the gel by crush-and-soak into TEN buffer (10 mM Tris-HCl pH 8.0, 1 mM EDTA pH 8.0, 300 mM NaCl) and recovered by EtOH precipitation.

Analytical denaturing polyacrylamide gels (0.75 × 100 × 80 mm) of the same composition were run for 1 h at 350 V constant voltage in 1x TBE, stained with SYBR Gold for 5 min in 1x TBE and imaged on an ImageQuant LAS 4000 (GE Healthcare).

Standard 5′-*O*-DMT/2′-*O*-TBDMS-protected 3′-β-cyanoethyl phosphoramidites were purchased from Sigma-Aldrich and ChemGenes. Isomeric 5′-*O*-DMT/3′-*O*-TBDMS-protected 2′-β-cyanoethyl phosphoramidites were purchased from ChemGenes. Phosphate CPG support (1000 Å) was purchased from Glen Research (USA). All other reagents for solid-phase synthesis were purchased from Sigma-Aldrich. HPLC purified oligonucleotides containing 2′,3′-cyclic phosphates or 3′ phosphates were purchased from ChemGenes and Merck, respectively. Primer sequences are detailed in Supplementary Data File 2.

## In vitro phosphatase assay

Oligonucleotides were 5′ labelled using T4 PNK (3′ phosphatase minus) (New England Biolabs) according to the manufacturer's instructions. In total, 200 fmol of labelled substrate were incubated in 100 mM Tris pH 7.5, 10 mM MgCl₂, 10 mM DTT and 50 mM KCl with the indicated amounts of recombinant *Dm*Angel, *Mm*ANGEL2 or *Hs*ANGEL2 prepared in a buffer containing 25 mM Tris pH 7.4, 0.5 mM EDTA, 10% glycerol, 3 mM β-mercaptoethanol and 500 mM NaCl. *Mm*ANGEL2 and *Hs*ANGEL2 reactions were incubated at 37 °C and *Dm*Angel reactions were incubated at 30 °C for 30 min and stopped by the addition of an equal volume of Gel Loading Buffer II (Thermo Fisher Scientific) and incubation at 70 °C for 10 min. Samples were loaded in 20% polyacrylamide gels, exposed to PhosphorImager screens and scanned in a Typhoon FLA7000 system. Quick CIP (New England Biolabs) and T4 PNK (New England Biolabs) control treatments were performed according to manufacturer's recommendations.

## Immunoprecipitation of DmAngel-FLAG and CG13850-FLAG

For *Dm*ANGEL-FLAG immunoprecipitation, embryos expressing *Dm*ANGEL or *Dm*ANGEL-FLAG were collected and transferred to vials with normal or heavy lysine as described in ref. 78. After 7 days, mitochondria were purified from larvae as described above. For CG13850-FLAG immunoprecipitation, larvae expressing CG13850, or CG13850-FLAG were grown in standard fly food and collected for mitochondria isolation 4 days after egg laying.

In total, 2 mg of mitochondria were lysed in 50 mM Tris-HCl pH 7.5, 150 mM NaCl, 6% digitonin with protease inhibitor cocktail (Roche). Samples were centrifuged at 20,000 × g for 15 min at 4 °C and the supernatant was incubated overnight with ANTI-FLAG M2 agarose beads (Merck) with rotation. Beads were pelleted at 5000 x g for 30 s and washed five times with 50 mM Tris-HCl pH 7.5, 150 mM NaCl. Beads containing the heavy and light labelled samples were pooled for proteomic analysis.

## In organello translation assays

Mitochondrial de novo translation was assayed as previously described[76]. In brief, 0.5 mg of mitochondria were incubated in translation buffer (100 mM mannitol, 10 mM sodium succinate, 80 mM KCl, 5 mM MgCl₂, 1 mM KH₂PO₄, 200 mM ATP, 5 mM GTP, 200 mM creatine phosphate, 6 mM creatine kinase, 100 μg/ml cycloheximide, 25 mM

HEPES pH 7.4 and 60 μg/ml of every amino acid except methionine) and 150 μCi of an $^{35}$S-labelled mix of methionine and cysteine, EXPRESS protein labelling mix easy-tag (Perkin Elmer), for 1 h at 37 °C. After labelling, equal amounts of total mitochondrial protein were separated on 16% or 17% SDS-PAGE gels, followed by staining with 1 g/l Coomassie Brilliant Blue in a 20% ethanol, 10% acetic acid solution. Gels were then destained, dried and exposed to a PhosphorImager screen.

### Sucrose gradient centrifugation analysis and ribosome profiling

Mitochondria from $DmAngel^{KO}$ larvae or $MmANGEL2^{KO}$ heart tissue were washed in ribosome buffer (50 mM KCl, 10 mM MgCl$_2$, 10 mM Tris pH 7.4). In total, 0.5 mg of mitochondria were lysed in 100 mM NH$_4$Cl, 10 mM MgCl$_2$, 1% Triton X-100, 30 mM Tris pH 7.6, and protease inhibitor cocktail, followed by centrifugation at 8000 × $g$ for 30 min at 4 °C to remove debris. The supernatant was loaded onto a 10–30% sucrose gradient in 10 mM MgCl$_2$, 100 mM NH$_4$Cl, 30 mM Tris pH 7.4 with protease inhibitor cocktail and RNase inhibitors. Gradients were centrifuged in a SWTi55 rotor at 39,000 × $g$ for 3 h at 4 °C. Twenty fractions were collected from each gradient. For western blot analysis, proteins in each fraction were precipitated and loaded on 12% polyacrylamide gels.

For ribosome profiling experiments, mitochondrial lysates were treated with 200U of RNase T1 for 30 min at room temperature, centrifuged and applied to a 10–30% sucrose gradient as described above. The monosome fractions were pooled and RNA was extracted using Direct-zol RNA micro prep kit (Zymo Research) following manufacturer's instructions and incorporating an on-column RNase-free DNase digestion to remove DNA. RNA was dephosphorylated and phosphorylated with T4 PNK. RNA sequencing libraries for ribosome profiling were prepared using the Illumina Small RNA Sample Prep Kit, with a size selection step for 15–90 nt RNAs. Initially, sequenced read pairs were merged with BBMerge (useoverlap = t pfilter = 1 mininsert = 18 mininsert0 = 18 trimnonoverlapping = t)[79] to remove non- overlapping sequences and trimmed of remaining linker sequence (AACTGTAGGCACCATCAAT) with BBDuk (ktrim = r mink = 4 k = 18 hdist = 1 tpe tbo) from the BBTools suite v38.79. Subtractive alignment was performed against *Drosophila* ncRNA sequences downloaded from Ensembl, including nuclear rRNA and tRNAs, with Bowtie2 (−fr)[80]. Unmapped reads were mapped to the *Drosophila* Ensembl v99 transcriptome sequences with Bowtie2 v2.4.1 (−very-sensitive−norc), with mitochondrial sequences modified to more accurately reflect the mature RNA transcripts present within the mitochondrion. In short, the singular annotations for bicistronic transcripts were merged (*mt:atp8/mt:atp6* and *mt:nd4/mt:nd4l*); 5′ and 3′ UTRs were added to select transcripts based on the observed positions of 5′ ends and poly(A) tails in RNASeq data (*mt:nd1, mt:nd3, mt:nd5, mt:co3, mt:cytb*); 30 nt poly(A) tails were added to the 3′ end of each mRNA and the large mitochondrial rRNA; a 5′ G was added to mt:tRNA:His-GTG and 3′ CCA added to all tRNAs to reflect the nontemplated additions added post-transcriptionally. Mitochondrial mRNA coverage bedgraph files were generated with BEDTools[81] v2.26.0 and normalised to total reads mapped to mitochondrial transcripts.

### Sample preparation for proteomic analysis

Proteomics analysis was performed essentially as previously described[63,82]. Mouse heart tissue was homogenised in 8 M urea, 50 mM ammonium bicarbonate, and 400 μm LoBind silica beads followed by sonication. In total, 25 μg of protein were digested with 0.5 μg trypsin (Promega) for 16 h at 37 °C, followed by the addition of 2.5 μl formic acid. Samples were cleaned on a C18 Hypersep plate and labelled with TMT-10plex reagents in random order, adding 100 μg TMT-reagent in 30 μl dry acetonitrile (ACN) to each digested sample resolubilised in 70 μl of 50 mM triethylammonium bicarbonate

(TEAB), and incubating at room temperature (RT) for 2 h. The labelling reaction was stopped for 15 min (RT) with 5% hydroxylamine before pooling. In total, 22 μg were stage tipped and dried before resuspended in 20 μl of 0.1% formic acid and 2% ACN.

Fly mitochondria isolated as described above and lysed with 8 M urea, 0.2% ProteaseMAX™ surfactant in 50 mM ammonium bicarbonate and sonicated. Proteins were reduced with dithiothreitol and alkylated with iodoacetamide, followed by digestion with 1.5 μg of trypsin for 16 h at 37 °C, and desalting and labelling as described above. The TMT-labelled tryptic peptides were fractionated on an XBridge C18 UPLC column (2.1 mm inner diameter × 250 mm, 2.5 μm particle size, Waters), and profiled with a linear gradient of 5–60% 20 mM ammonium hydroxide in ACN (pH 10.0) over 48 min (flow rate of 200 μl/min). Fractions were collected at 30 s intervals into a 96-well plate and combined in 12 samples concatenating 8–8 fractions representing peak peptide elution.

For SILAF-IPs, agarose beads were supplemented with 30 μl of 4 M urea in 100 mM Tris-HCl, pH 7.5 and 1 μl of 0.5 μg/μl Lys-C and incubated at 27 °C for 30 min, before collection of the supernatant. The beads were washed in 1 M Urea, 50 mM Tris-HCl pH 7.5, 1 mM DTT and with 100 mM Tris-HCl pH 7.5 twice. Digestion in the eluent was continued over night at RT adding 1 μl of 0.5 μg/μl Lys-C with shaking at 400 rpm, while the agarose beads were supplemented with 50 μl of 100 mM Tris-HCl and further digested with 1 μl of 0.5 μg/μl Lys-C overnight at RT with shaking at 400 rpm. Following over-night digestion at RT, the peptide solutions were alkylated with 0.6 M iodoacetamide in 50 mM of AmBic, pH 8 for 30 min in the dark. The digestion was stopped with 10 μl of formic, cleaned on a C-18 HyperSep plate (Thermo Scientific) and dried in a Speedvac. Label-free IPs were processed as SILAF-IPs, except for the addition of 0.5 μg trypsin overnight.

### Liquid chromatography-tandem mass spectrometry data acquisition

TMT-labelled peptides were reconstituted in solvent A and ~2 μg samples (3/12 μl) injected on a 50 cm long EASY-Spray C18 column (Thermo Fisher Scientific) connected to an Ultimate 3000 nanoUPLC system (Thermo Fisher Scientific) using a 90 min long gradient: 4–26% of solvent B (98% acetonitrile, 0.1% FA) in 90 min, 26–95% in 5 min, and 95% of solvent B for 5 min at a flow rate of 300 nl/min. Mass spectra were acquired on a Q Exactive HF hybrid quadrupole orbitrap mass spectrometer (Thermo Fisher Scientific) ranging from $m/z$ 375 to 1500 at a resolution of $R$ = 120,000 (at $m/z$ 200) targeting 5 × 106 ions for maximum injection time of 80 ms, followed by data-dependent higher-energy collisional dissociation (HCD) fragmentations of precursor ions with a charge state 2+ to 8+, using 45 s dynamic exclusion. The tandem mass spectra of the top 18 precursor ions were acquired with a resolution of $R$ = 60,000, targeting 2 × 105 ions for maximum injection time of 54 ms, setting quadrupole isolation width to 1.4 Th and normalised collision energy to 33%.

SILAF IP peptides were reconstituted in 15 μl solvent A and 5 μl injected on a 50 cm long EASY-Spray C18 column (Thermo Fisher Scientific) connected to an EASY-nLC 1000 HPLC system (Thermo Fisher Scientific) using a 60 min long gradient: 4–25% of solvent B (98% acetonitrile, 0.1% FA) in 50 min, 25–40% in 10 min, 40–95% in 3 min, and 95% of solvent B for 5 min at a flow rate of 300 nl/min. Mass spectra were acquired on an Orbitrap Fusion Tribrid mass spectrometer (Thermo Fisher Scientific) ranging from $m/z$ 375 to 1500 at a resolution of $R$ = 120,000 (at $m/z$ 200) targeting 4 × 105 ions for maximum injection time of 50 ms, followed by data-dependent HCD fragmentations of precursor ions with a charge state 2+ to 7+, using 30 s dynamic exclusion. The tandem mass spectra were acquired with a resolution of $R$ = 30,000 in 2 s cycle time, targeting 5 × 104 ions for maximum injection time of 100 ms, setting quadrupole isolation width to 1.6 Th and normalised collision energy to 30%.

Label-free IP peptides were measured as TMT-labelled peptides, except for 30 s dynamic exclusion and a normalised collision energy of 28%.

## Proteomics data analysis

TMT proteomics data were mapped with Proteome Discoverer software (Thermo Fisher Scientific) version 2.5 against a UniProt *D. melanogaster* (downloaded October 2019 with 13,788 entries) and *Mus musculus* canonical database at default settings. SILAF proteomics data was mapped in MaxQuant v.1.6.3.4[83] at default settings with Lys0 or Lys6 as labels and LysC/P as protease against the canonical and isoform sequences of the fruit fly proteome (UP000000803_7227 from UniProt, downloaded in September 2018 with 21,939 entries). Normalised expression values, or normalised heavy/light ratios were further analysed in R v4.1.1. DmAngel knock-out fly proteomic data on fractionated mitochondria was normalised to median intensities of mitochondrial proteins to account for differences in organellar enrichment. A limma moderated *t*-test was applied for statistical testing. MitoXplorer v2[84] protein sets were used as an annotated mitochondrial protein library. OXPHOS complex annotations were manually curated in-house. Proteomics data have been deposited to the ProteomeXchange[85] Consortium.

## Reporting summary

Further information on research design is available in the Nature Research Reporting Summary linked to this article.

## Data availability

The data that support this study are available from the corresponding authors upon reasonable request. The ribo-profile data sets generated in this study have been deposited to GEO with accession number GSE189750. MS proteomics data generated in this study have been deposited to the ProteomeXchange Consortium via the PRIDE[86] partner repository with the dataset identifier PXD033394. Datasets used in this study are MitoXplorer v2, FASTA files from UniProt (*D. melanogaster* in October 2019, *M. musculus* in September 2018) and *Drosophila* Ensembl v99 transcriptome sequences. Source data are provided with this paper.

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

## Acknowledgements

The authors would like to thank Prof. Clara Green, University of Texas, for kind gifts of NOCT KO tissue samples. The *daughterless*GAL4 line was a kind gift of Prof. Linda Partridge, Max Planck Institute for the Biology of Ageing. Protein identification and quantification were carried out by the Proteomics Biomedicum core facility, KI (https://ki.se/en/mbb/proteomics-biomedicum). Computations using Quandenser were performed on resources provided by Swedish National Infrastructure for Computing (SNIC) through Uppsala Multidisciplinary Centre for Advanced Computational Science (UPPMAX) under project SNIC 2019-8-175 and uppstore2018188. Funding that supported this research was provided by: Swedish Research Council VR2016-02179 (A.W.), Novo-Nordisk Foundation NNF18OC0032200 (A.W.), European Research Council 715009 (A.W.), Knut & Alice Wallenberg Foundation KAW2019.0109 (A.W.), Knut and Alice Wallenberg Foundation: WAF 2017, KAW 2018.0080, Karolinska Institutet and the Max Planck Institute (J.R.), KAW 2016.0087 (K.P.), EMBO Postdoctoral Fellowship ALTF 1011-2020 (C.S.), National Health and Medical Research Council (A.F. and O.R.), Australian Research Council (A.F. and O.R.).

## Author contributions

Conceptualisation: P.C., C.F., and A.W.; Methodology: P.C., Y.K., O.R., K.P., A.F., J.R., C.F., and A.W.; Investigation: P.C., J.C.G., S.F.P., F.A.S., M.S., S.J.S., I.L., H.S., C.S., and R.W.; Visualisation: P.C., J.C.G., and C.F.; Funding acquisition: Y.K., O.R., K.P., A.F., J.R., and A.W.; Project administration: P.C., C.F., and A.W.; Supervision: C.F. and A.W.; Writing—original draft: P.C., C.F., and A.W.; Writing—review and editing: all authors.

## Funding

## Competing interests

The authors declare no competing interests.
