## [Peer Review File · Nature Communications]

REVIEWER COMMENTS

Reviewer #1 (Remarks to the Author):

Thank you very much for sending me this manuscript from Clemente et al. to review. It is well written and structured and I found it fascinating. Congratulations to the authors for identifying this novel mitochondrial protein, Angel2. I am convinced by much of the data. For example, I particularly liked the work around the MPAT assays shown in Fig3. I don't think there is any doubt that Angel2 is a mito matrix protein and that it can dephosphorylate immature mt-mRNA that, unusually, carries a 3' phosphate. The authors also show that this function is important for maturing primary transcripts at sites that are not marked by mt-tRNAs. Finally, they also report that CG13850 is a drosophila ortholog of a FASTK family protein. Where I would caution the authors is over their claim that they have now shown how non-canonical processing occurs in mitochondria. I think the data is great to support the role of Angel2 in this process and that it is highly likely that FASTK proteins are also involved but to make this claim I would like to see such unprocessed RNA being correctly cleaved and matured, ready for polyadenylation, in vitro. That said, if the authors toned this claim down a bit I think the paper is excellent. I have a variety of points and comments the authors may wish to consider:

What is perhaps remarkable is how little resultant phenotype there is in the KO mouse, particularly as both ATP6/8 and ND5 require this form of Angel2 maturation. As the MmAngel2 KO is perfectly viable for at least 16 wks it would have been really interesting to know how tissues other than the heart are able to compensate for the loss. Do they still have non-canonically processed species that lack polyA extensions due to the loss of the phosphatase, or is there another phosphatase activity that can compensate? What does this data say about the importance of mt-mRNA polyadenylation, particularly for ATP8/6? There is no evidence of CV assembly defects in mouse heart (supp fig2j). Therefore, is there any associated translation defect in the KO mouse? In Fig 4 I couldn't see any evidence of expts that purport to MmAngel2. This is quite important, as the claim in the figure legend is that the loss of Angel2 can affect translation in mammals as well as drosophila. Supp Fig5a is suggested to show subtle translation effects in mouse heart and liver. I'm afraid I couldn't see anything that in my opinion would not fall into experimental variation. I think if there was clear evidence of mito translation modulation then one might expect to see a more obvious phenotype. In Dm, however, it is more clear. It is interesting to note that in man the loss of a termination codon in ATP6/8 leads to translation dependent decay. With the Angel2 KO, this transcript is definitely not low at steady state and there is no clear decrease in the level of CV, suggesting there is no such mechanism occurring. Perhaps 3' phosphate shields the transcript from the normal translation dependent decay mechanism? Why are there three bands in the Angel2-protC expt in Fig1c? Why is ND2 protein level so low in the KO fly in Fig 4.

Reviewer #2 (Remarks to the Author):

The manuscript by Clemente et. al. attempts to shed light on non-canonical processing of mitochondrial transcripts that aren't punctuated by tRNAs. Authors suggest that members of FAST kinase domain containing protein family cleave such unconventional transcript junctions, leaving a 3'-phosphate that inhibits proper polyadenylation and translation of the upstream transcript. Importantly, they show that carbon catabolite repressor 4 domain-containing family member ANGEL2 is required to rescue transcripts that have undergone non-canonical processing by hydrolyzing such 3'-phosphates. The authors also show that ANGEL2 is localized to mitochondria and its deletion causes respiratory chain deficiencies in

Drosophila and mouse models. Overall, the study is intriguing, novel, and timely, and the manuscript is succinct and well written; however, the authors need to perform additional experiments to justify some of their claims. Furthermore, the presentation of data in some figures may benefit from additional editing. My specific comments are listed below:

Major Comments:

1. The nuclease activity of FASTK needs to be elaborated with in-vitro transcribed RNA substrate(s). This is important since the details of how FASTK would recognize and excise the non-canonical junctions is unknown.

An in-vitro demonstration that FASTK cleavage will leave behind 3' phosphates using RtcB assay, and that ANGEL2 can rescue this defect are needed to support the model shown in Fig. 5f. This is required to prove that this activity is independent of RNaseP/ELAC2 as shown in 5f.

2. The paper introduces the idea that KO of DmANGEL and Angel2 causes respiratory chain deficiency, but it seems more like an afterthought as it's not given much attention in the primary figures and doesn't fit in with the rest of the paper's main findings. Additionally, there appears to be a large difference in the necessity of the proteins in mice vs flies, and no conjecture is given as to why, despite the focus on their functional homology.

3. Previous studies have shown that non-canonical mt-mRNA processing is not essential to mammalian survival. How do the authors explain Angel2 knockout's lethal phenotype if non-canonical mitochondrial mRNA processing is not essential to mammalian cell survivability?

4. What are the relative expression levels of ANGEL1 and ANGEL2 in mouse? The authors should evaluate potential functional compensation by ANGEL1 in ANGEL2 KO mice.

5. ANGEL1 is mentioned to be localized to the outer mitochondrial membrane in mouse. ANGEL2 is predicted to have better functional homology with DmANGEL based on localization to mitochondrial matrix. Have the authors evaluated ANGEL1 KO in mice, or double knock out of ANGEL proteins in mice?

6. It would be informative to add a section detailing if any non-mitochondrial functions of ANGEL2 are known. While DmANGEL is shown to be important for development, the authors do not conclusively show that it is in fact the respiratory defects that leads to lethality in flies (Mice seem to survive fine).

7. The discussion section almost entirely focuses on mammalian non-canonical processing, only mentioning the Drosophila equivalent, DmANGEL once, however, their results are much more focused on DmANGEL, leading to a disconnect. It seems odd that they would focus their attention almost entirely on ANGEL2 when their results showed a much greater physiological impact of DmANGEL KO.

Additional comments:

1. Fig. 1A: The authors should use an immunostaining-based method to demonstrate the mitochondrial localization of intrinsic ANGEL proteins in different cell types. Mitochondrial localization is not evident from the current set of images. Higher magnification images are required to show mitochondrial localization. All merged images should have a zoomed in panel, which would allow for more accurate determination of overlap. Also, scale-bars are missing from the images.

2. Fig. 1B: Low level of hsANGEL2:ProteinC expression in the nucleus does not match with confocal images (Fig. 1A), wherein hsANGEL2 has a very strong nuclear signal. Authors should also explain the differences in Protein C migration in different fractions. Immunoblot for intrinsic hsANGEL2 should also be provided.

3. Fig. 1C: The inferences regarding the sub-mitochondrial localization of ANGEL1 and 2 are

vaguely explained and need to be supported by additional direct experiments. Also, the authors have drawn functional homology between hsANGEL2 and DmAngel only on the basis of their 'co-localization' and have not backed this claim by any functional assay.

0. Supplementary Fig 1C: Why are there multiple FLAG bands at different MW in all the three lanes?

1. Fig 1D and Supplementary Fig 2I: mitochondrial dysfunction about complex V is claimed but not supported by any data.

2. Supplementary Fig 1G & H: X-axis labels are hard to interpret. Knockdown and rescue should be in the same graph, with proper controls.

3. The formatting of the northern blot in Figure 2A (specifically the alternating KO/WT) makes it harder than it should be to draw conclusions from the figure, a grouped approach like that seen in Figure 2E would be much more readable.

4. The RT-PCR data in Figure 2C doesn't include Nd6, which is one of the few differences seen from the Northern Blot quantifications in Figure 2A—it seems like an odd choice to not include it.

5. Supplementary Fig 2A: As ANGEL2 knockout is not very robust at the RNA levels, the knockout should be confirmed at the protein level by immunoblot.

0. In panels 3A and 3B, wildtype animals consistently display very faint bands for transcripts processed in non-canonical fashion, compared to bands for canonical transcripts. Why does such a trend exist? [For instance, in 4a, compare WT bands (-/CIP/PNK) of ATP8/6 vs Co3 or Nd5 vs Cytb]. The non-canonical does show strong band in KOs after CIP treatment.

1. The first section of the results "CCR4-family..." contains a fair amount of background information that would possibly be better introduced in the introduction instead of the results section.

2. There are also minor spacing issues, "non-canonical" being the most inconsistent problem (spacing with the dash is inconsistent throughout the paper).

Reviewer #3 (Remarks to the Author):

In this manuscript, the authors used IP-MS to identify DmANGEL interacting proteins in mitochondria, where they found CG13850, a member of the FASTK protein family, interact with DmANGEL. The deletion of CG13850 prevented the cleavage during non-canonical RNA processing. The proteomics work along with other experiments, while out of my expertise, are generally well designed and performed. The data interpretation and presentation are also clear and straightforward. My only minor concern is why only LysC but not trypsin as well was used as digestion enzyme for proteomic analysis, as trypsin is largely accepted in the proteomics community and generate the most comprehensive peptide IDs. The authors may reasoned that they are only quantifying isotope-labeled Lys peptides, but further cleavage after arginine can help shorten the Lys peptides and benefit for identification.

COMMENTS TO THE REVIEWERS:

Reviewer #1 (Remarks to the Author):

Thank you very much for sending me this manuscript from Clemente et al. to review. It is well written and structured and I found it fascinating. Congratulations to the authors for identifying this novel mitochondrial protein, Angel2. I am convinced by much of the data. For example, I particularly liked the work around the MPAT assays shown in Fig3. I don't think there is any doubt that Angel2 is a mito matrix protein and that it can dephosphorylate immature mt-mRNA that, unusually, carries a 3' phosphate. The authors also show that this function is important for maturing primary transcripts at sites that are not marked by mt-tRNAs. Finally, they also report that CG13850 is a drosophila ortholog of a FASTK family protein. Where I would caution the authors is over their claim that they have now shown how non-canonical processing occurs in mitochondria. I think the data is great to support the role of Angel2 in this process and that it is highly likely that FASTK proteins are also involved but to make this claim I would like to see such unprocessed RNA being correctly cleaved and matured, ready for polyadenylation, in vitro. That said, if the authors toned this claim down a bit I think the paper is excellent. I have a variety of points and comments the authors may wish to consider:

We thank the reviewer for their positive feedback. We appreciate that the reviewer agrees with our interpretation of the data concerning ANGEL2. We are delighted to have finally identified the function of these ANGELS and place them into the underexplored mechanism of non-canonical processing in mitochondria. We agree with the reviewer that more functional data is required for CG13850 and its mammalian homologs, and although a growing body of evidence links this family of enzymes to non-canonical processing, we do not demonstrate cleavage activity directly. We therefore took the advice from the reviewer and restructured the manuscript. However, we do believe that our results, together with previous work, explains the mechanism of non-canonical processing in metazoa.

What is perhaps remarkable is how little resultant phenotype there is in the KO mouse, particularly as both ATP6/8 and ND5 require this form of Angel2 maturation. As the MmAngel2 KO is perfectly viable for at least 16 wks it would have been really interesting to know how tissues other than the heart are able to compensate for the loss. Do they still have non-canonically processed species that lack polyA extensions due to the loss of the phosphatase, or is there another phosphatase activity that can compensate?

This is an interesting thought and to exclude the presence of other phosphatases in other tissues, we have performed MPAT analysis on several additional tissues, with the same result. We now demonstrate the accumulation of terminal phosphates in heart, liver, kidney, and skeletal muscle of ANGEL2 KO

animals. (See figure S5). Based on these results, we propose that ANGEL2 activity is required in most major tissues, if not all. Nevertheless, we agree with the reviewer that ANGEL2 KO mice have a surprisingly mild phenotype, especially when considering KO models of other mitochondrial gene expression factors. In this respect, we would argue that FASTKD4 or FASTKD5 KO mouse models would be affected much more, which we would like to investigate, if the next grant cycle permits us. ANGEL2 KO mice do, however, present with reduced complex I activity already at 16 weeks, and reduced complex I activity is a hallmark of, for instance, Parkinson's disease, as well as the natural ageing process, and we are therefore currently performing more detailed phenotyping at different time points, now that we have identified the molecular mechanism. We hope to be able to present this data in the future.

What does this data say about the importance of mt-mRNA polyadenylation, particularly for ATP8/6 ? There is no evidence of CV assembly defects in mouse heart (supp fig2j). Therefore, is there any associated translation defect in the KO mouse ? In Fig 4 I couldn't see any evidence of expts that purport to MmAngel2. This is quite important, as the claim in the figure legend is that the loss of Angel2 can affect translation in mammals as well as drosophila. Supp Fig5a is suggested to show subtle translation effects in mouse heart and liver. I'm afraid I couldn't see anything that in my opinion would not fall into experimental variation. I think if there was clear evidence of mito translation modulation then one might expect to see a more obvious phenotype. In Dm, however, it is more clear.

The reviewer is correct in that the translation defect is much more pronounced in the fly than in the mouse heart. At this point, we do not have a clear explanation for this. Although the gene organisation between fly and mouse differs slightly, only *Atp6* of the affected transcripts requires polyadenylation to complete a stop codon, and the ribosome footprint analysis we performed suggests stalling at the 3' terminus of *atp6* in flies. However, the flies do have a more prominent effect on mt-tRNA stability, with mt-tRNA^{Cys} and mt-tRNA^{Tyr} severely reduced in *Drosophila*. We do not know why these tRNAs are affected, but we have seen an isolated effect on mt-tRNA^{Cys} levels before in a fly model deficient of the mitochondrial poly(A) polymerase MTPAP (Bratic et al 2016 PLoS Genetics). We have now clarified this effect on *Drosophila* mt-tRNAs in the manuscript.

Regarding the *in organello* experiments in mouse ANGEL2 KO hearts, we agree that the effect on translation is mild, but we have performed multiple repeats of this experiment and consistently observe an increase in ND5 peptide throughout all our experiments. The reason is most likely due to the increased transcript levels, rather than due to lack of polyadenylation. However, our proteomics data from mouse hearts (Figure 1g and S2k) does suggest a stability effect on complex I, potentially also driven by the reduced *mt:nd6* transcript levels (Figure 2b). Surprisingly, though, we did not observe reduced translation of ND6. For ATP6 the interpretation is more difficult as ATP6 separates together

with COX2 and COX3, making it difficult to distinguish. Why ATPase activity is not affected in the same way in the mouse as in the fly is not entirely clear to us, and we will continue to study how these tail-less ATP8/6 transcripts are translated and released. To reflect this, we have changed the heading of the figure legend of figure 4 accordingly and added a section to the discussion.

It is interesting to note that in man the loss of a termination codon in ATP6/8 leads to translation dependent decay. With the Angel2 KO, this transcript is definitely not low at steady state and there is no clear decrease in the level of CV, suggesting there is no such mechanism occurring. Perhaps 3' phosphate shields the transcript from the normal translation dependent decay mechanism?

The effect on ATP6 translation and complex V activity is indeed surprising. However, we do not anticipate a translation dependent decay. Temperley et al reported the rapid degradation of a mutant *atp8/6* transcript in a patient with a microdeletion at the 3' terminus of *ATP8/6* (Temperley et al 2003 HMG). This mutation removed the terminal stop codon of *atp6* but retained polyadenylation, resulting in a poly-lysine synthesis. We assume that the poly-lysine signal is a major contributor to the decay of these transcripts. In contrast, the *atp6* transcripts described here lack a functional stop codon, as well as polyadenylation signal, and thus do not encode for a poly-lysine. We previously demonstrated that in the absence of polyadenylation transcripts are stable and translated but are trimmed at their 3' termini (Batic et al. 2016 PLoS Genetics). We did not perform MPAT assays in that particular study, so we can only speculate how such transcripts would resolve on MPAT assays, but we do observe an increased signal below the zero position of *atp8/6* in ANGEL2 KO mice and *atp8/6* and *nd6* in *DmANGEL* KO flies, consistent with trimmed transcripts. Most transcripts, however, seem to be full length, phosphorylated, and non-polyadenylated, indeed suggesting protection. *In vitro* kinetic experiments with recombinant PNPase and SUV3 and RNA templates with various terminal phosphate configurations might be informative.

Why are there three bands in the Angel2-protC expt in Fig1c ?

We believe that that the largest molecular weight band is unprocessed ANGEL2 prior to mitochondrial import, while the middle band present in the total and cytosol fractions represents an unspecific band due to cross-reactivity, as we see the same band in extracts from fibroblasts expressing non-tagged ANGEL2 (See Figure 1 of this response letter).

Figure 1. Subcellular localisation of ANGEL2.

Why is ND2 protein level so low in the KO fly in Fig 4.

In general, there seems to be some derangement in mitochondrial translation in the fly in response to the deletion of DmANGEL. As the reviewer indicates, ND2 seems to be decreased, while several other bands become more prominent or more diffuse, as is the case for ND1. This was one of the reasons why we performed ribosome profiling, of which we present the data in our manuscript. At this point, we are unable to fully explain why and how translation is further affected in the fly but might also be due to changes in tRNA levels. We hope to be able to fully understand the response in the future.

Reviewer #2 (Remarks to the Author):

The manuscript by Clemente et. al. attempts to shed light on non-canonical processing of mitochondrial transcripts that aren't punctuated by tRNAs. Authors suggest that members of FAST kinase domain containing protein family cleave such unconventional transcript junctions, leaving a 3'-phosphate that inhibits proper polyadenylation and translation of the upstream transcript. Importantly, they show that carbon catabolite repressor 4 domain-containing family member ANGEL2 is required to rescue transcripts that have undergone non-canonical processing by hydrolyzing such 3'-phosphates. The authors also show that ANGEL2 is localized to mitochondria and its deletion causes respiratory chain deficiencies in drosophila and mouse models. Overall, the study is intriguing, novel, and timely, and the manuscript is succinct and well written; however, the authors need to perform additional experiments to justify some of their claims. Furthermore, the presentation of data in some figures may benefit from additional editing. My specific comments are listed below:

Major Comments:

1. The nuclease activity of FASTK needs to be elaborated with in-vitro transcribed RNA substrate(s). This is important since the details of how FASTK would recognize and excise the non-canonical junctions is unknown.

An in-vitro demonstration that FASTK cleavage will leave behind 3' phosphates using RtcB assay, and that ANGEL2 can rescue this defect are needed to support the model shown in Fig. 5f. This is required to prove that this activity is independent of RNaseP/ELAC2 as shown in 5f.

We thank the reviewer for their positive assessment of our manuscript, and we agree that the details of how FASTK recognises gene junctions involved in non-canonical processing is not known.

This manuscript is the first report of the functional targets of ANGEL2 and DmANGEL, as well as the first demonstration of the presence of terminal phosphates because of nuclease activity inside mitochondria. Furthermore, we demonstrate this in two animal models, as well as reproduce the previously reported phosphatase activity of both ANGEL2 and *DmANGEL in vitro* (see below). We feel that a full demonstration of ANGEL2, *DmANGEL*, as well as FASTK family members in two species *in vivo* and *in vitro* is beyond the scope of this manuscript. After all, the mechanism of canonical processing was proposed 40 years ago, and the details are still not fully solved. It would be naïve of us to suggest that we can solve non-canonical processing in a single publication. As suggested by reviewer 1, we have therefore restructured the manuscript.

However, we disagree that we need to demonstrate that RNaseP/ELAC2 are not involved in non-canonical processing. Numerous publications demonstrate that non-canonical processing is not affected upon disruption of either of these enzyme complexes and having to re-demonstrate this here is, in our view, excessive.

2. The paper introduces the idea that KO of DmANGEL and Angel2 causes respiratory chain deficiency, but it seems more like an afterthought as it's not given much attention in the primary figures and doesn't fit in with the rest of the paper's main findings. Additionally, there appears to be a large difference in the necessity of the proteins in mice vs flies, and no conjecture is given as to why, despite the focus on their functional homology.

Measuring respiratory chain function is a routine investigation, when dealing with factors thought to affect mitochondrial gene expression. In fact, the clinic where one of us is an active MD, and where these measurements were done, performs several respiratory chain enzyme activity and respiration measurements every week as one of the first steps during the diagnosis of patients with suspected metabolic disease. The information gained is not only a demonstration of physiological relevance but can also direct the investigator towards a specific part of mitochondrial function. For instance, a combined complex I, III, IV, and V deficiency, but normal complex II activity, might indicate a defect in mitochondrial gene expression, as complex II is the only fully nuclear-encoded complex, while an isolated respiratory chain complex defect points to a defect in a complex subunit of assembly factor. Measuring respiratory chain function is therefore a standard procedure in the field and was one of the first experiments performed. In our case, the data demonstrates that loss of DmANGEL and ANGEL2 leads to a respiratory chain dysfunction. The effect in the mouse is indeed mild, but mild complex I defects are a hallmark in some neurodegenerations and in ageing tissues. Further investigations will therefore be of interest. The data was therefore not an afterthought but important and strengthens our conclusion of the function of DmANGEL and ANGEL2. As discussed with reviewer 1, it might result in an interesting discussion and while we might not be able to answer all the questions the data evokes, it hopefully contributes to our understanding of mitochondrial function in a physiological context.

We agree with the reviewer that our previous discussion was relatively short, and we realise now that there are several points that need to be further clarified for the reader. We have therefore expanded the discussion, including our interpretation of the different phenotypes.

3. Previous studies have shown that non-canonical mt-mRNA processing is not essential to mammalian survival. How do the authors explain Angel2 knockout's lethal phenotype if non-canonical mitochondrial mRNA processing is not essential to mammalian cell survivability?

We are not aware of any previous report demonstrating the involvement of a protein specific to non-canonical processing. Thus, we are not aware of any report suggesting that non-canonical processing inside mitochondria is not essential for mammalian survival. In fact, we would argue that we are the first to demonstrate an exclusive mechanism of non-canonical processing. Several cell culture models disrupting FASTKD genes alone or in combination have been published, but none demonstrated a

specific function that is exclusive to non-canonical processing, and since even cells without mtDNA can be maintained in culture, we do not classify this as surviving *in vivo*. It will be interesting to study FASTKD KO mouse models, as to date, no clinical phenotype has been associated with FASTKD4 or 5.

Nevertheless, we assume the reviewer must have confused the two animal models we used, as *Angel2*^{KO} mice are viable, while loss of ANGEL in the fly is indeed larva lethal. We have now included a section in the discussion regarding this.

4. What are the relative expression levels of ANGEL1 and ANGEL2 in mouse? The authors should evaluate potential functional compensation by ANGEL1 in ANGEL2 KO mice.

Transcript expression levels in multiple tissues and cell types can be readily found on multiple databases, and we are not sure what this would add to the current manuscript. Additionally, we have not been able to identify antibodies that have been demonstrated to recognise endogenous ANGEL1 or ANGEL2. The commercially available antibody used in figure 1b only worked upon overexpression of ANGEL1, and we unsuccessfully attempted to generate our own antibodies against ANGEL1 and ANGEL2.

Nevertheless, we extensively demonstrate compartmentalisation of ANGEL1 and ANGEL2, and are therefore not convinced that a functional compensation is possible. After all, the deletion of ANGEL2 and its *Drosophila* ortholog DmANGEL leads to the accumulation of terminal phosphates on mitochondrial transcripts, arguing that the presence of ANGEL1 is not sufficient to prevent this accumulation. As mentioned to reviewer 1, we now show the presence of terminal phosphates in additional tissues. Additionally, Pinto et al. demonstrated ANGEL1 phosphatase activity was only comparable to ANGEL2 at 3000-4000 fold molar concentrations, questioning its physiological relevance as a phosphatase (Pinto et al. 2020 Science).

5. ANGEL1 is mentioned to be localized to the outer mitochondrial membrane in mouse. ANGEL2 is predicted to have better functional homology with DmANGEL based on localization to mitochondrial matrix. Have the authors evaluated ANGEL1 KO in mice, or double knock out of ANGEL proteins in mice?

Our claim that ANGEL2 is the ortholog of DmANGEL is not purely based on their respective localisation, but also on the fact that the deletion of ANGEL2 or DmANGEL leads to the accumulation of terminal phosphates on the same mitochondrial transcripts. In our view, this strongly supports our conclusion that ANGEL2 and DmANGEL are indeed orthologs. Furthermore, we now demonstrate 2' and 2'3'-cP phosphatase activity for both ANGEL2 and DmANGEL *in vitro* (Figure 4C).

We are not sure whether the reviewer here really suggests that we would need to additionally characterise ANGEL1 and ANGEL1/2 double KO mouse models. We hope that was not the intention.

As discussed, given the current information we have on the localisation of ANGEL1 and ANGEL2 we see no point in trying to interpret a combined phenotype from two distinct defects. After all, compartmentalisation is one of the fundamental principles in eukaryotes. Furthermore, since the loss of ANGEL2 leads to the accumulation of terminal phosphates, it is unlikely that ANGEL1 can compensate for the deletion of ANGEL2.

However, we have indeed generated ANGEL1 KO mice, which are viable, with no effect on mitochondrial transcripts and no mitochondrial dysfunction. Moreover, ANGEL1 does not share the interactome with ANGEL2, and ANGEL1 KO hearts do not accumulate terminal phosphates on mitochondrial transcripts, as determined by our MPAT analysis. Nevertheless, the investigation of this model is still ongoing and will be presented elsewhere. We have not generated double ANGEL KO models, as we do not anticipate that they are part of the same biological mechanism.

6. It would be informative to add a section detailing if any non-mitochondrial functions of ANGEL2 are known. While DmANGEL is shown to be important for development, the authors do not conclusively show that it is in fact the respiratory defects that leads to lethality in flies (Mice seem to survive fine).

Several publications previously suggested non-mitochondrial functions for ANGEL2. However, their data is difficult to interpret due to their experimental design. A nuclear localisation of ANGEL2 was previously reported by the Lykke-Andersen group. However, the authors used N-terminally tagged constructs to demonstrate localisation, which would have blocked the N-terminal mitochondrial targeting signal, preventing mitochondrial localisation. These authors additionally use an antibody generated against ANGEL2; however, validation of this antibody is not shown, and it fails to recognise endogenous ANGEL2 in the mitochondrial matrix. Likewise, the report by Pinto and colleagues (Pinto et al Science 2020) used an N-terminally trimmed ANGEL2 construct, again preventing mitochondrial localisation. Thus, previous reports either obscured or removed the mitochondrial targeting sequence, which can lead to cytosolic or nuclear localisation, which is well known in the field. In contrast, our results corroborate previous, unbiased observations (Rhee et al Science 2013, Antonicka et al Cell Metab 2020). Additionally, in contrast to these reports, we here demonstrate the targets of ANGEL2 and DmANGEL, and present their physiological roles. We have now expanded the discussion to include these points.

It is often impossible to conclusively demonstrate what is the ultimate cause of death in a biological system. The respiratory chain defect we observe in our fly model is consistent with defects we observe in other models of mitochondrial gene expression. It is therefore reasonable to conclude that the observed OXPHOS deficiency has a major impact on larvae survival. However, we do not know why flies are more affected than mice, and clearly more work is required, and at this point, we can only speculate.

7. The discussion section almost entirely focuses on mammalian non-canonical processing, only mentioning the *Drosophila* equivalent, DmANGEL once, however, their results are much more focused on DmANGEL, leading to a disconnect. It seems odd that they would focus their attention almost entirely on ANGEL2 when their results showed a much greater physiological impact of DmANGEL KO.

We appreciate that the discussion was kept short, and we have now expanded on several points. We have also emphasised the point that we demonstrate that DmANGEL and ANGEL2 are orthologs and the discussion around their function is therefore interchangeable unless stated otherwise.

Additional comments:

1. Fig. 1A: The authors should use an immunostaining-based method to demonstrate the mitochondrial localization of intrinsic ANGEL proteins in different cell types. Mitochondrial localization is not evident from the current set of images. Higher magnification images are required to show mitochondrial localization. All merged images should have a zoomed in panel, which would allow for more accurate determination of overlap. Also, scale-bars are missing from the images.

We do not agree that we need to further demonstrate ANGEL2 and DmANGEL localisation in different cell types. Which cell types would the reviewer suggest? And how many? We have tested multiple commercially available antibodies, and none are able to detect endogenous ANGEL2 levels. Only one antibody was able to detect ANGEL1 upon overexpression, which we have used in Figure 1b. Additionally, we attempted to generate our own antibodies from recombinant fly, mouse, and human protein, but again, these did not work. A growing number of publications demonstrate a mitochondrial localisation by various methods, including unbiased proteomic analysis of the mitochondrial proteome, as well as subcellular proximity reports. We feel that these reports, in combination with our localisation, subcellular and sub-mitochondrial fractionation, experiments, as well as the effect on mitochondrial transcripts demonstrates a mitochondrial localisation. We now present zoomed in sections. We apologise for not adding scale-bars.

2. Fig. 1B: Low level of hsANGEL2:ProteinC expression in the nucleus does not match with confocal images (Fig. 1A), wherein hsANGEL2 has a very strong nuclear signal. Authors should also explain the differences in Protein C migration in different fractions. Immunoblot for intrinsic hsANGEL2 should also be provided.

We believe that this is a consequence of the overexpression, and subcellular fractionation procedure. As mentioned, we were unable to generate intrinsic antibodies. Regarding the migration, please see our response to reviewer 1.

The mitochondrial localisation of ANGEL2 is supported by several publications investigating the mitochondrial proteome. We are aware of two publications suggesting otherwise, but as discussed

above, we believe these reports are misleading due to the constructs used. We much more suggest that the nuclear localisation of ANGEL2 in our confocal images is a consequence of overexpression. Re-examination of our images suggest that the nuclear localisation is indeed a rare event. Nevertheless, whether ANGEL2 indeed has an additional function outside of mitochondria will require further, much more careful, investigations.

3. Fig. 1C: The inferences regarding the sub-mitochondrial localization of ANGEL1 and 2 are vaguely explained and need to be supported by additional direct experiments. Also, the authors have drawn functional homology between hsANGEL2 and DmAngel only on the basis of their 'co-localization' and have not backed this claim by any functional assay.

We respectfully disagree with the reviewer. We demonstrate mitochondrial and submitochondrial localisation in multiple ways, using approaches fully recognised in the field. Our conclusion of our experiments is supported by proteomic analysis of the mitochondrial matrix (Rhee et al Science 2013, Antonicka et al Cell Metab 2020). Furthermore, a mitochondrial localisation is also supported in MitoCarta (Rath et al NAR 2020). Additionally, as discussed, a non-mitochondrial localisation has not yet been convincingly demonstrated.

We also disagree with the comment that our functional homology between ANGEL2 and DmANGEL is only on the basis of co-localisation. Figure 3 clearly demonstrates functional homology beyond just co-localisation. Additionally, we have now expressed recombinant ANGEL2 and DmANGEL and demonstrate phosphatase activity *in vitro* for both. This is now added to the manuscript.

4. Supplementary Fig 1C: Why are there multiple FLAG bands at different MW in all the three lanes?

This band is unspecific in flies, as it is also present in the non-flag tagged controls. Please see our response to reviewer 1.

5. Fig 1D and Supplementary Fig 2I: mitochondrial dysfunction about complex V is claimed but not supported by any data.

The reviewer is correct that Figure 1d only measures respiration (complexes I-IV) but not complex V (ATPase) activity. This is because isolated complex V activity cannot be measured in our setup. However, we show complex V in-gel activity in Figure S2I, clearly demonstrating hydrolysis activity of the F1 subunit of ATPase in a non-assembled complex (asterisk) and is consistent with complex V defect reported by others (Mourier et al. 2014, HMG 23:2580, Smet et al. 2009 Electrophoresis 30, Zhang et al. 2004 BBRC 322:565, Bratic et al. 2011 PLoS Genet).

6. Supplementary Fig 1G & H: X-axis labels are hard to interpret. Knockdown and rescue should be in the same graph, with proper controls.

We assume the reviewer refers to figure S2g and S2h. Also, we have not used any knockdown fly lines for DmANGEL in this manuscript. The use of a legend to refer to colours within a graph are common and having labels on the x-axis would be double labelling. We chose to make two graphs, as the flies and their genotypes in these crosses, are very different. The principle of fly genetics is to use balancer chromosomes to maintain heterozygous fly stocks. In these balancer chromosomes the gene order of large chromosomal regions has been scrambled to such a degree that recombination is prevented. Additionally, these chromosomes contain easily identifiable markers, such as eye colour or wing shape, and are often homozygous lethal, allowing for the easy maintenance of heterozygous stocks without the need of genotyping every fly. This is a well-established system in *Drosophila* and goes back to the middle of the last century.

In our case, we “balanced” the *DmANGEL*^{KO} locus to the *CyO* balancer of chromosome 2, which has curly wings as phenotypic marker and is homozygous lethal. For Figure S2g in each experiment 100 eggs from the following cross were picked and the number of hatching flies recorded (control experiments are performed using wild type *White*^{dahomey} (*wDah*):

$$\begin{array}{c} DmAngel^{K^{\circ}} \\ CyO \text{ _____} \end{array} \times \begin{array}{c} \overline{DmAngel^{K^{\circ}}} \\ CyO \end{array}$$

In this case we only obtained offspring with curly wings, meaning that homozygous KO *DmAngel* flies are not viable, as these would not contain the *CyO* balancer and therefore have straight wings. Homozygous *CyO* is embryonic lethal. According to Mendelian genetics this will result in 50% heterozygous flies, which is in line with the control experiment, as not all flies hatch under normal conditions.

For figure S2h we crossed

$$\begin{array}{c} DmAngel^{K^{\circ}} \\ CyO \text{ _____} \end{array} \times \begin{array}{c} DmAngel^{Rescu} \\ \text{'DmAngel}^{Rescue} \text{X} \end{array} \times \begin{array}{c} DmAngel^{K^{\circ}} \\ CyO \text{ _____} \end{array} \times \begin{array}{c} \overline{DmAngel^{Rescue}} \\ \text{'DmAngel}^{Rescue} \end{array}$$

where *DmAngel*^{KO} is on chromosome 2 and the rescue construct (*DmAngel*^{Rescue}) integrated on chromosome 3 and homozygous viable. In this case, 25% of flies would have straight wings, if the homozygous *DmAngel*^{KO} locus can be rescued by the rescue construct on chromosome 3. Since not all eggs will hatch a 10% rescue is considered a rescue, especially as the homozygous KO has no hatching flies at all. Given that we used 3 separate lines (*DmAngel*^{Rescue}, *DmAngel*^{Rescue-FLAG}, *DmAngel*^{E121A}), we chose to only show the relevant genotypes for simplicity.

However, we did notice that we mislabelled the graph and have replaced it with a new one, better reflecting the genotypes used.

7. The formatting of the northern blot in Figure 2A (specifically the alternating KO/WT) makes it harder than it should be to draw conclusions from the figure, a grouped approach like that seen in Figure 2E would be much more readable.

We respectfully suggest that this is semantics and reviewers in the past wanted this arrangement, instead of grouped. Figure 2b is the quantification of 2a for clarity. The reason why figure 2e is grouped is because we have two independent KO fly lines, making alternate loading more difficult.

8. The RT-PCR data in Figure 2C doesn't include Nd6, which is one of the few differences seen from the Northern Blot quantifications in Figure 2A—it seems like an odd choice to not include it.

We assure the reviewer that excluding *nd6* in the RT-PCR was a deliberate choice. *Nd6* mRNA fully overlaps with *nd5*, which is encoded on the opposite strand. Thus, Nd6 should never be quantified by qRT-PCR, as the cDNA pool generated by random hexamers does not allow us to distinguish between *nd6* and *nd5* transcripts. In contrast, primers used to quantify *nd5* are outside of this overlap region. Northern blots do not have this limitation since we routinely use single stranded probes to detect *nd6* transcripts, as mentioned in the M&Ms.

9. Supplementary Fig 2A: As ANGEL2 knockout is not very robust at the RNA levels, the knockout should be confirmed at the protein level by immunoblot.

As shown in figure S2a, the KO of ANGEL2 involves the deletion of 32 nucleotides in exon 3 of ANGEL2, leading to a downstream frame shift and premature stop codon. The primers used to quantify *angel2* transcript levels are outside of this deletion and it is reasonable that transcription of the deleted locus can still occur. The reduction in transcript levels is rather a sign of nonsense mediated decay. As mentioned above, we do not have a functional antibody for ANGEL2.

10. In panels 3A and 3B, wildtype animals consistently display very faint bands for transcripts processed in non-canonical fashion, compared to bands for canonical transcripts. Why does such a trend exist? [For instance, in 4a, compare WT bands(-/CIP/PNK) of ATP8/6 vs Co3 or Nd5 vs Cytb]. The non-canonical does show strong band in KOs after CIP treatment.

The images are taken by phosphorimaging and are dependent on the exposure. To prevent the signal from non-polyadenylated transcripts from the CIP/PNK treatment to bleed the entire bottom half of the gel, we had to adjust the exposure accordingly. Since *mt:co3* and *mt:cytb* do not contain non-polyadenylated transcripts, we could adjust the exposure to visualise the poly(A) tail.

11. The first section of the results "CCR4-family..." contains a fair amount of background information that would possibly be better introduced in the introduction instead of the results section.

We feel that mentioning the CCR4 family in the introduction would not be logical as we had not yet identified CCR4C, CCR4D or CCR4E as factors of interest. We therefore prefer to leave it where it is but have included more discussion regarding previous work on the CCR4 family.

12. There are also minor spacing issues, “non-canonical” being the most inconsistent problem (spacing with the dash is inconsistent throughout the paper).

The two spacing issues in line 62 and 63 have been fixed. We apologies for these.

Reviewer #3 (Remarks to the Author):

In this manuscript, the authors used IP-MS to identify DmANGEL interacting proteins in mitochondria, where they found CG13850, a member of the FASTK protein family, interact with DmANGEL. The deletion of CG13850 prevented the cleavage during non-canonical RNA processing. The proteomics work along with other experiments, while out of my expertise, are generally well designed and performed. The data interpretation and presentation are also clear and straightforward. My only minor concern is why only LysC but not trypsin as well was used as digestion enzyme for proteomic analysis, as trypsin is largely accepted in the proteomics community and generate the most comprehensive peptide IDs. The authors may reasoned that they are only quantifying isotope-labeled Lys peptides, but further cleavage after arginine can help shorten the Lys peptides and benefit for identification.

We thank the reviewer for their feedback. We previously developed an *in vivo* labelling technique, we termed SILAF (stable-isotope labelling of amino acids in flies), which is based on the traditional SILAC labelling in cells, and uses a holidic food source to directly feed to flies (Schober et al. 2020, 2021). SILAC labelling in cells usually uses both heavy lysine and arginine, but our studies demonstrated that arginine is partially metabolised in the fly, making MS data difficult to interpret. We therefore only used heavy lysine, but since trypsin preferentially digests after arginine residues, MS data would contain both heavy and light signals from the same protein. We therefore only use LysC to avoid this artefact. We have discussed this in our previous work (Schober et al. 2021). We have also now included a detailed proteomic description in the materials and methods.

REVIEWERS' COMMENTS

Reviewer #1 (Remarks to the Author):

I'd like to thank the authors for their thoughtful and complete responses to my comments. In my opinion, this is an excellent piece of work and is now acceptable for publication.

Reviewer #2 (Remarks to the Author):

This is a thoroughly revised manuscript. I would like to thank the authors to have invested the time and resources to generate new data, and a more refined discussion to make this a compelling piece of work. I have no further concerns.